# D-ARL: A Distribution-Matched Asynchronous Reinforcement Learning Framework for Language Reasoning

**Yinqi Bai** [1]  **Xialiang Tong** [2]  **Jie Wang** [1 †]  **Hongyu Liu** [1]  **Longdi Pan** [1]  **Jiashuo Li** [1]  **Zehao Wang** [1]  **Jianye Hao** [3]
**Mingxuan Yuan** [2]  **Feng Wu** [1]

## Abstract

Asynchronous reinforcement learning (RL) has shown notable success in accelerating the post-training of large language models (LLMs). However, its decoupled data generation and training paradigm introduces a fundamental distributional mismatch between data generated by stale behavior policies and current policy, leading to unstable training and degraded performance. To address this challenge, we propose D-ARL, a **D**istribution-matched **A**synchronous **R**einforcement **L**earning framework that selects high-quality asynchronous samples whose distributions are well aligned with the current policy for policy optimization. Specifically, D-ARL maintains a replay buffer that collects samples from the most recent $K$ behavior policies and proposes a variance-guided metric to select distribution-matched data. During training, D-ARL introduces a multi-behavior policy optimization algorithm to leverage the multi-source nature of the selected samples for policy update. Experiments on six widely used reasoning benchmarks show that D-ARL outperforms state-of-the-art asynchronous methods, achieving an average improvement of 6.4% in reasoning performance and 34.7% in sample efficiency. We open-source our code at `https://github.com/YinqiBai962/D-ARL`.

## 1. Introduction

Reinforcement learning (RL) has become a core technique for the post-training of large language models (LLMs)

(Zhang et al., 2025). Recently, most existing RL algorithms rely on synchronous training pipelines, where data generation and policy optimization proceed in a sequential manner (Shao et al., 2024; Schulman et al., 2017; Konda & Tsitsiklis, 1999). Such synchronization often leads to inefficient resource utilization and severe training bottlenecks, especially in large-scale LLM training where policy rollout and optimization incur substantial computational overhead (Uehara et al., 2022). To alleviate these limitations, asynchronous reinforcement learning frameworks have been proposed to decouple data generation from policy updates, thereby enabling parallel execution (Noukhovitch et al., 2025b). By relaxing synchronization constraints, asynchronous RL significantly improves training efficiency, making it an appealing solution for LLM post-training (Lu et al., 2025).

Existing asynchronous RL methods always collect samples generated by the most recent behavior policies for policy updates, as shown in Figure 1 (Chen et al., 2025; Hilton et al., 2022). However, due to the inherent asynchrony between stale behavior policies and current policy, these methods suffer from a fundamental distributional mismatch between off-policy data and current policy, leading to unstable training and degraded performance (Noukhovitch et al., 2025a).

To address this challenge, we propose a novel Distribution-matched Asynchronous Reinforcement Learning (D-ARL) framework. The D-ARL is motivated by two key observations: (1) Asynchronous data whose distribution matches the current policy is particularly effective for training; (2) The behavior policies at different time steps can generate distribution-matched samples. Motivated by these observations, D-ARL proposes a distribution-matched sample selection mechanism as shown in Figure 1. It maintains a replay buffer containing samples from the most recent $K$ behavior policies, and defines an off-policy degree metric to measure the distributional discrepancy between asynchronous samples and a theoretically optimal behavior policy. Based on this metric, D-ARL selects a batch of samples with the smallest degree from the replay buffer. Furthermore, to leverage the multi-source nature of the selected samples drawn from $K$ distinct behavior policies, D-ARL proposes a multi-behavior policy optimization algorithm. This algo-

[1] MoE Key Laboratory of Brain-inspired Intelligent Perception and Cognition, University of Science and Technology of China, China [2] Huawei Technologies Co., Ltd., China [3] College of Intelligence and Computing, Tianjin University, China . Correspondence to: Jie Wang <jiewangx@ustc.edu.cn>.

*Proceedings of the 43 $^{rd}$ International Conference on Machine Learning*, Seoul, South Korea. PMLR 306, 2026. Copyright 2026 by the author(s).

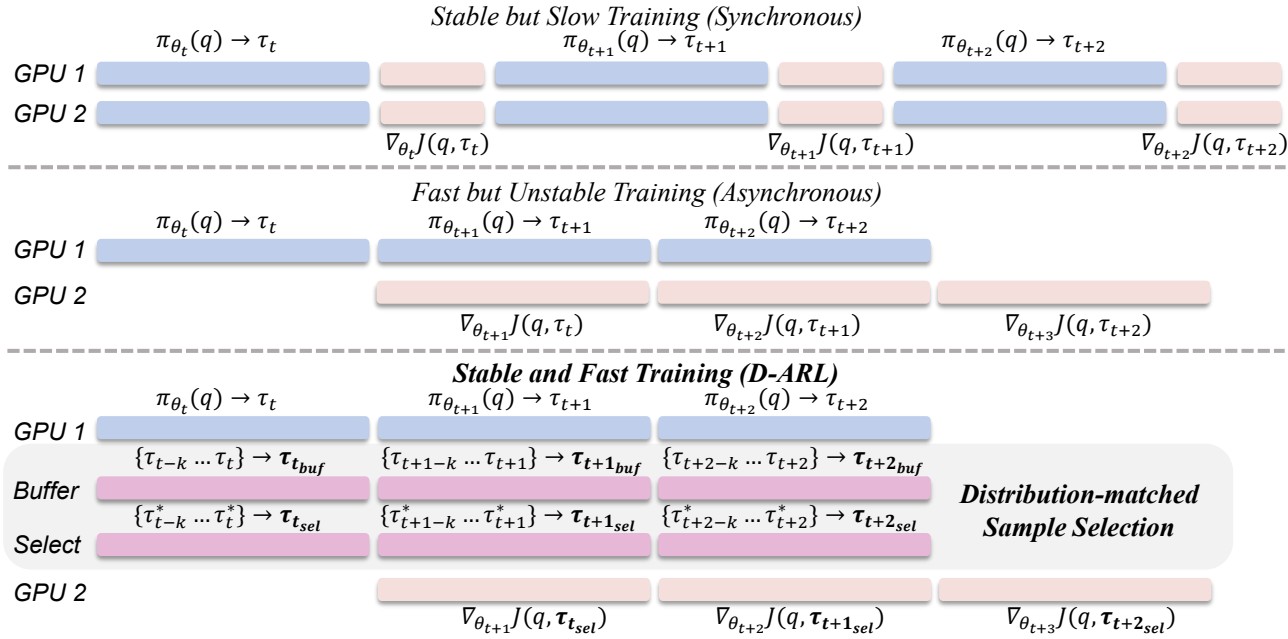

Figure 1. Our D-ARL framework effectively combines the advantages of both asynchronous and synchronous training, achieving stable and fast training through distribution-matched sample selection.

rithm introduces optimal policy-level weighting for each policy domain, effectively aggregating domain information and thus improving training performance.

Experiments on six widely used mathematical and code reasoning benchmarks demonstrate that D-ARL consistently outperforms previous state-of-the-art (SOTA) asynchronous RL methods in both reasoning performance and sample efficiency. Across all experiments, we use Qwen3-1.7B and Qwen3-4B as the backbone models. Specifically, D-ARL achieves an average improvement of 6.4% in reasoning performance over the SOTA baseline. Moreover, D-ARL attains the same accuracy with only 65.3% of the training steps required by the SOTA method, highlighting the superior training efficiency of our approach.

We summarize our main contributions as follows: (1) To the best of our knowledge, D-ARL is *the first* asynchronous RL framework to identify and leverage distribution-matched data for high-performance asynchronous training. (2) We propose a novel multi-behavior policy optimization algorithm that can effectively aggregate information from heterogeneous asynchronous data sources for stable policy updates. (3) Experiments on diverse benchmarks demonstrate that our method significantly outperforms the SOTA baselines in reasoning performance and training efficiency.

## 2. Background

**Asynchronous Reinforcement Learning** Traditional RL methods adopt synchronous training pipelines, where data collection and policy optimization are tightly coupled

and executed sequentially (Shao et al., 2024; Schulman et al., 2017). This design limits training throughput and underutilizes computational resources. To improve efficiency and scalability, recent asynchronous RL approaches (Noukhovitch et al., 2025a; Lu et al., 2025; Fu et al., 2025) decouple rollout generation from policy updates, enabling parallel training and significantly accelerating RL post-training (Figure 1).

**Distribution Shift in Asynchronous RL** While asynchronous RL substantially improves training efficiency, it inevitably introduces off-policy data, i.e., rollout trajectories generated by outdated behavior policies. This results in a distribution shift between the collected data and the current policy, which can destabilize learning and degrade policy performance (Lu et al., 2025; Meng et al., 2023). Consequently, effectively addressing this distribution mismatch has emerged as a central challenge in asynchronous RL.

**Off-Policy Optimization and Importance Sampling (IS)** In asynchronous reinforcement learning, off-policy optimization methods have been developed to leverage the off-policy data for policy updates (Meng et al., 2023). A principled approach for off-policy methods is Importance Sampling (IS) (Hanna et al., 2021). Specifically, given a trajectory $\tau$ from a behavior policy $\pi_q$ and a reward function $R$, the expected return $J$ of a target policy $\pi_p$ is defined as:

$$J = \mathbb{E}_{\tau \sim \pi_p}[R(\tau)] = \mathbb{E}_{\tau \sim \pi_q}\left[\frac{\pi_p(\tau)}{\pi_q(\tau)}R(\tau)\right], \quad (1)$$

where $\frac{\pi_p(\tau)}{\pi_q(\tau)}$ is the importance ratio that reweights off-policy samples to match the target policy distribution. However,

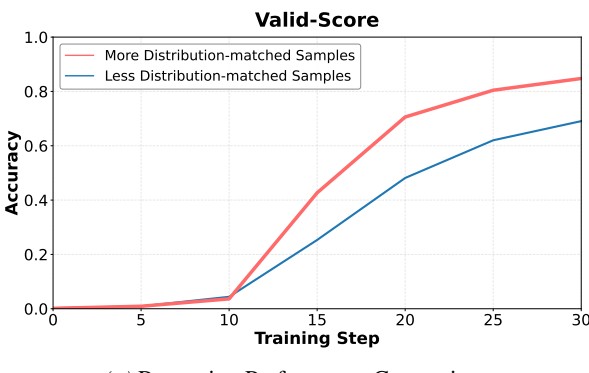

*(a)* Reasoning Performance Comparison

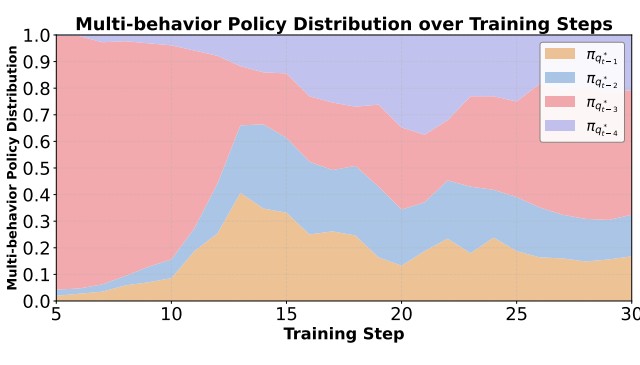

*(b)* Probability Distribution Evolution

*Figure 2.* (a) demonstrates that, under the same sample number, samples with better distributional matching yield stronger reasoning performance. (b) shows that even highly off-policy behavior policies generate high-quality training samples with low off-policy degree.

directly applying IS can introduce large variance in the return estimates, especially when the behavior and target policies differ significantly (Espeholt et al., 2018).

## 3. Motivating Results

**Distribution-Matched Samples Lead to Better Training**
Traditional asynchronous RL methods typically use all samples generated by the most recent behavior policy for training. However, not all of these samples contribute equally to learning. In particular, we observe that asynchronous data whose distribution closely matches the current policy is especially effective for policy optimization. To quantify this, we propose a metric (see Definition 4.2) that evaluates the off-policy degree of each sample and uses it to select distribution-matched samples for training. As shown in Figure 2a, samples that are better aligned with the current policy provide greater reasoning performance, indicating the effectiveness of the distribution-matched samples.

**Past Behavior Policies Generate High-quality Samples**
While traditional approaches assume that samples generated by the most recent behavior policy exhibit the lowest off-policy degree, we find that samples from earlier behavior policies can also remain highly aligned with the current policy. Specifically, we analyze the source distribution of top-ranked samples with the smallest off-policy degrees among the most recent four behavior policies. As shown in Figure 2b, only about 18.1% of these samples originate from the most recent behavior policy for Qwen3-1.7B on the GSM8K benchmark, indicating that historical behavior policies can also produce high-quality samples.

## 4. Method

In this section, we present a detailed description of the Distribution-Matched Asynchronous Reinforcement Learning (D-ARL) framework (see Figure 3). We first introduce a distribution-matched sample selection mechanism that iden-

tifies high-quality asynchronous data for training. We then propose a multi-behavior off-policy optimization method to effectively leverage the selected data from multiple behavior policies for policy optimization. The detailed training algorithm is illustrated in Algorithm 1.

### 4.1. Distribution-matched Sample Selection

Motivated by the observation that highly off-policy behavior policies can still generate high-quality asynchronous samples (see Figure 2b), we maintain a replay buffer that stores samples generated by the previous $K$ policies. However, not all buffered samples are equally beneficial for learning (see Figure 2a), and naively using the entire buffer can impair training efficiency. Therefore, we propose a principled mechanism for selecting high-quality asynchronous data. Our core idea is to first characterize the optimal behavior distribution $\pi_{q^*}$ that minimizes the variance of the Importance Sampling (IS) estimator, and then select samples from the replay buffer whose distributions best match $\pi_{q^*}$. We provide more implementation details in Appendix E.2.

**Minimum-Variance Behavior Policy** In Asynchronous RL, IS provides an unbiased estimator for off-policy optimization. However, it often leads to a high-variance estimator, which negatively impacts training stability (Espeholt et al., 2018). Therefore, controlling and minimizing the variance of the IS estimator is a central objective in asynchronous and off-policy learning. Let $\pi_p$ and $\pi_q$ denote the target and behavior policy distributions, respectively, and let $R$ be the reward associated with trajectory $\tau$. Given i.i.d. samples $\{\tau_i\}_{i=1}^n$ drawn from $\pi_q$, the estimator of expected return under IS (see equation 1) is defined as

$$\hat{J}_{\pi_q} = \frac{1}{n} \sum_{i=1}^{n} \frac{\pi_p(\tau_i)}{\pi_q(\tau_i)} R(\tau_i). \quad (2)$$

To find a behavior distribution $\pi_{\pi_q}$ that minimizes the variance of $\hat{J}_{\pi_q}$, we propose the following Minimum-Variance Behavior Policy Theorem.

## 1. Distribution-matched Sample Selection

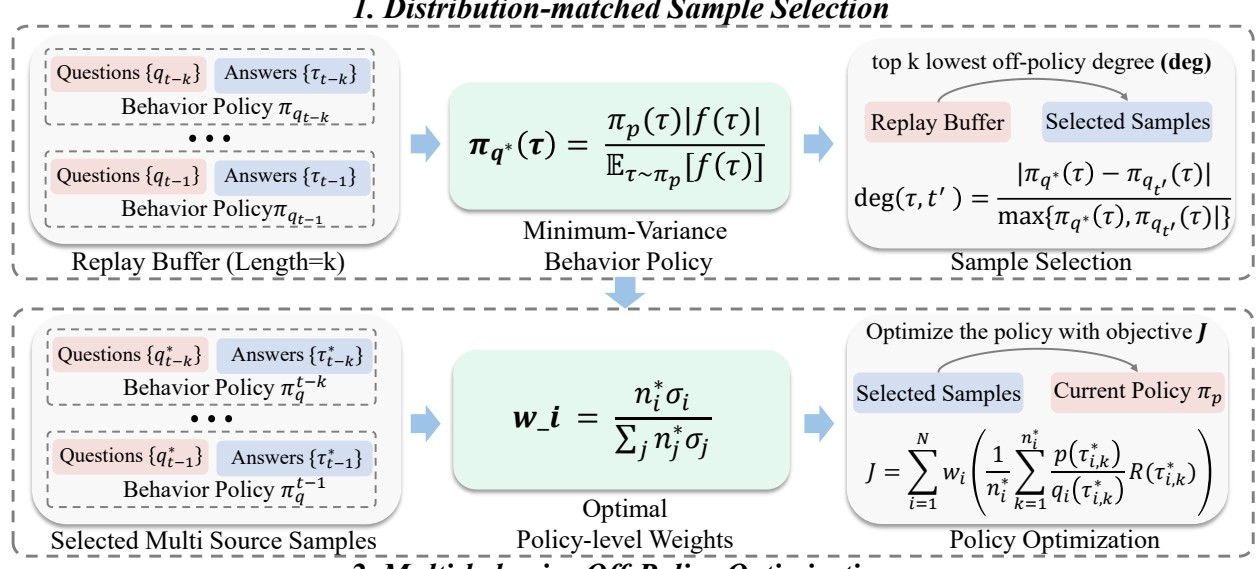

## 2. Multi-behavior Off-Policy Optimization

*Figure 3.* Illustration of the D-ARL framework. Specifically, D-ARL introduces a two-stage framework. First, it proposes a distribution-matched sample selection mechanism to identify and prioritize high-quality asynchronous samples. Subsequently, building upon these selected samples, D-ARL proposes a multi-behavior policy optimization method to update the policy.

**Theorem 4.1.** *Among all behavior policies that yield an unbiased estimator $\hat{J}_{\pi_q}$, the variance of it is minimized when*

$$\pi_{q^*}(\tau) = \frac{\pi_p(\tau)|R(\tau)|}{\mathbb{E}_{\tau \sim \pi_p}[R(\tau)]}. \quad (3)$$

The proof of Theorem 4.1 is provided in Appendix A.1. However, in Equation 3, the term $\mathbb{E}_{\tau \sim \pi_p}[R(\tau)]$ represents an expectation over the unknown target policy and cannot be computed exactly in practice. To make $\pi_{q^*}$ operational, we approximate this term by the average of returns observed in past time steps, thereby leveraging historical information to estimate the expectation under the target policy. Due to limited space, more details about the optimal behavior policy are provided in Appendix E.2.1.

**Sample Selection** Based on Theorem 4.1, we propose an off-policy degree criterion for each sample in the buffer to quantify its alignment with the optimal behavior policy.

**Definition 4.2** (Off-Policy Degree). Let $\tau$ be a trajectory sampled from a behavior policy $\pi_q$. The off-policy degree of $\tau$ is then defined as

$$\deg(\tau) = \frac{|\pi_{q^*}(\tau) - \pi_q(\tau)|}{\max\{\pi_{q^*}(\tau), \pi_q(\tau)\}}. \quad (4)$$

The off-policy degree quantifies how closely a sample aligns with the optimal behavior policy $\pi_{q^*}$. Accordingly, we compute the off-policy degree for all samples in the replay buffer and select the top $k$ samples with the lowest degree for policy optimization. These selected samples are most closely aligned with $\pi_{q^*}$ and therefore contribute effectively to stable asynchronous training. Additional details of the off-policy degree are provided in Appendix E.2.2.

### 4.2. Multi-behavior Policy Optimization

Traditional off-policy optimization methods typically rely on samples generated from the most recent behavior policy (Noukhovitch et al., 2025a). In contrast, our selected samples originate from multiple behavior policies. Consequently, we model the policy optimization process as a multi-behavior off-policy problem and propose a corresponding multi-behavior policy optimization method. Specifically, we first present our off-policy modeling. Let $\{\pi_{q_i}\}_{i=1}^N$ denote $N$ behavior policies, each generating $n_i^*$ samples $\{\tau_{i,k}\}_{k=1}^{n_i^*}$. We define the return estimator as

$$\hat{J}_{\text{DARL}} = \sum_{i=1}^N w_i \left( \frac{1}{n_i^*} \sum_{k=1}^{n_i^*} \frac{\pi_p(\tau_{i,k})}{\pi_{q_i}(\tau_{i,k})} R(\tau_{i,k}) \right), \quad (5)$$

where $w_i$ denotes the weight assigned to the $i$-th behavior policy. These weights allow us to aggregate information from multiple behavior policies in a principled manner, giving higher importance to policies whose samples are more aligned with the target policy. More implementation details are provided in Appendix E.3.

**Optimal Policy-level Weights** We present the following theorem to provide the optimal form of the weights $w_i$.

**Theorem 4.3.** *To minimize the variance of the return estimator, the optimal policy-level weights are given by*

$$w_i = \frac{n_i^* \sigma_i}{\sum_{j=1}^N n_j^* \sigma_j}, \quad (6)$$

*where* $\sigma_i = \frac{1}{\text{Var}_{\tau \sim \pi_{q_i}}\left[ \frac{\pi_p(\tau)}{\pi_{q_i}(\tau)} R(\tau) \right]}$.

The proof of Theorem 4.3 is provided in Appendix A.2. The optimal weights $w_i$ in Theorem 4.3 reflect the relative contribution of each behavior policy $\pi_{q_i}$ to the overall return estimation. Specifically, $w_i$ is proportional to both the number of samples $n_i^*$ generated by policy $\pi_{q_i}$ and the variance $\sigma_i$ of the importance-weighted returns. This formulation has an intuitive interpretation: a policy that generates more samples or produces lower-variance returns is assigned a larger weight, ensuring that its information is appropriately emphasized in the aggregated estimator. By incorporating these weights, the estimator $\hat{J}_{\mathrm{DARL}}$ achieves minimum variance across all behavior policies, thereby improving the stability and efficiency of off-policy optimization.

**Policy Optimization** Based on Theorem 4.3, we could compute the expected return estimator $\hat{J}_{\mathrm{DARL}}$. In our experiments, we adopt the Group Relative Policy Optimization (GRPO) algorithm (Shao et al., 2024). Here, the return $R(\tau)$ of a trajectory $\tau$ corresponds to the cumulative reward along the trajectory, which is further adjusted by the group advantage to capture the collective contribution of related trajectories or policies. After computing $\hat{J}_{\mathrm{DARL}}$, we update the current policy using gradient-based optimization with the selected asynchronous samples. Although we use GRPO in our experiments, the proposed DARL framework is general and can also be applied to other policy-gradient-based RL algorithms, such as PPO (Schulman et al., 2017).

# 5. Experiments

In this section, we conduct extensive experiments to evaluate the effectiveness of our proposed D-ARL. The experimental evaluation consists of four main parts: **Experiment 1.** To demonstrate the superior performance of D-ARL in reasoning performance. **Experiment 2.** To demonstrate the superior performance of our method in training efficiency. **Experiment 3.** We conduct carefully designed ablation experiments to provide further insight into the contributions of individual components of D-ARL. **Experiment 4.** We provide a deep analysis of the selected high-quality asynchronous samples from D-ARL.

**Tasks and Datasets** We evaluate D-ARL on six widely used public reasoning benchmarks covering both mathematical reasoning and code generation tasks. For mathematical reasoning, we adopt GSM8K (Cobbe et al., 2021), LightEval (Hendrycks et al., 2021), AIME24 (Zhang & Math-AI, 2024), and MATH-500 (Lightman et al., 2023). For code generation, we employ Livecodebench (Jain et al., 2024) and HumanEval (Chen et al., 2021) as evaluation benchmarks. For more details, please refer to Appendix D.

**Experimental Setup** Throughout all experiments, we adopt the VERL (Sheng et al., 2025) framework as the unified backbone for both training and evaluation, and use Qwen3-

1.7B and Qwen3-4B as the base models. For mathematical reasoning benchmarks GSM8K and LightEval, we adopt the official training split, while for MATH-500 and AIME24, training is performed using the mathematical subset of the Guru-RL-92K dataset (Cheng et al., 2025). For code generation tasks, models are trained on code samples from the Guru-RL-92K dataset and evaluated on LiveCodeBench and HumanEval following their standard evaluation protocols. Across all experiments, we use a batch size of 256 and train all methods for one epoch. The length of the replay buffer is defined as 4. All experiments are conducted on a single node equipped with four NVIDIA A100 GPUs (80GB memory each). More details about the experimental setup are provided in Appendix E.1.

**Evaluation Metrics** Throughout all experiments, we adopt *reasoning performance* and *training efficiency* as the primary evaluation metrics. Reasoning performance is defined as the proportion of correctly answered questions on a held-out test set. To assess training efficiency, we consider both *sample efficiency* and *training time*. Sample efficiency is measured by the number of training steps required to reach a target accuracy, while training time denotes the corresponding wall-clock time. Together, these metrics enable a comprehensive evaluation of both the training performance and efficiency of the comparative methods.

**Baselines and Hyperparameters** We compare our D-ARL with four baseline methods, including the standard synchronous GRPO algorithm and three state-of-the-art asynchronous baselines as follows. For more implementation details of the baselines, please refer to Appendix E.1.

- **GRPO** (Shao et al., 2024) is a standard on-policy Group Relative Policy Optimization algorithm. It normalizes rewards within response groups to compute relative advantage signals, serving as our primary synchronous baseline.

- **ARLHF** (Noukhovitch et al., 2025a) is an asynchronous variant of GRPO that performs policy updates by leveraging samples collected from the most recent training step, together with explicit IS correction to utilize these asynchronous samples.

- **Decoupled** (Hilton et al., 2022) introduces an auxiliary proximal policy to regulate policy updates when training with stale samples. By decomposing the IS ratio into two stages, this method improves optimization stability under significant policy drift.

- **CISPO** (Chen et al., 2025) constrains the importance sampling ratio within an asymmetric interval around 1, controlled by lower and upper bounds $\epsilon_{\mathrm{low}}^{\mathrm{IS}}$ and $\epsilon_{\mathrm{high}}^{\mathrm{IS}}$. Compared to symmetric clipping strategies, this formulation provides finer-grained control over the trust region during policy optimization.

*Table 1.* Accuracy comparison of D-ARL and baseline methods on six benchmarks using the Qwen3-1.7B and Qwen3-4B models. The results demonstrate that D-ARL consistently outperforms all baselines in terms of reasoning performance.

| Qwen3-1.7B | Mathematical | | | | Coding | | Average |
|---|---|---|---|---|---|---|---|
| | GSM8K | LightEval | MATH-500 | AIME24 | LiveCodeBench | HumanEval | |
| **Method** | Acc (%) ↑ | Acc (%) ↑ | Acc (%) ↑ | Acc (%) ↑ | Acc (%) ↑ | Acc (%) ↑ | Acc (%) ↑ |
| GRPO | 74.6 | 48.5 | 72.2 | 13.3 | 43.5 | 65.9 | 53.0 |
| ARLHF | 69.1 | 48.1 | 72.2 | 16.7 | 44.8 | 63.6 | 52.4 |
| CISPO | 79.9 | 59.8 | 76.3 | 16.7 | 49.8 | 70.1 | 58.8 |
| Decoupled | 64.4 | 47.5 | 70.4 | 26.7 | 47.7 | 71.3 | 54.7 |
| D-ARL (Ours) | **84.8** | **65.7** | **78.7** | **26.7** | **54.4** | **81.1** | **65.2** |

| Qwen3-4B | Mathematical | | | | Coding | | Average |
|---|---|---|---|---|---|---|---|
| | GSM8K | LightEval | MATH-500 | AIME24 | LiveCodeBench | HumanEval | |
| **Method** | Acc (%) ↑ | Acc (%) ↑ | Acc (%) ↑ | Acc (%) ↑ | Acc (%) ↑ | Acc (%) ↑ | Acc (%) ↑ |
| GRPO | 88.8 | 57.4 | 81.3 | 30.0 | 47.7 | 76.8 | 63.7 |
| ARLHF | 88.6 | 56.3 | 80.1 | 26.7 | 48.5 | 78.1 | 63.1 |
| CISPO | 92.0 | 70.3 | 84.5 | 33.3 | 51.1 | 81.1 | 68.7 |
| Decoupled | 89.7 | 56.7 | 81.1 | 33.3 | 48.5 | 76.2 | 64.2 |
| D-ARL (Ours) | **94.0** | **73.5** | **87.3** | **40.0** | **60.7** | **90.2** | **74.3** |

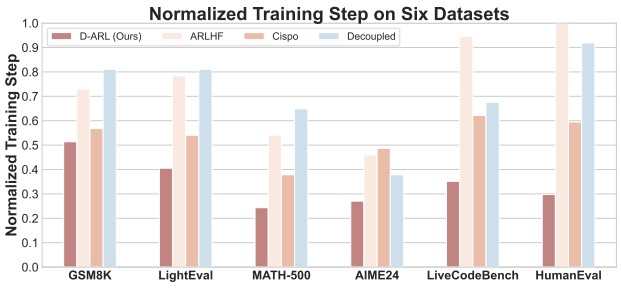

*(a)* Normalized Training Step for Qwen3-1.7B

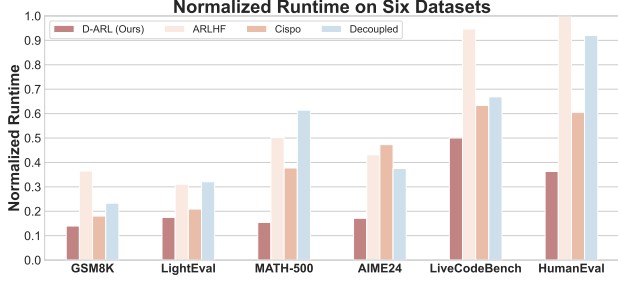

*(b)* Normalized Training Time for Qwen3-1.7B

*Figure 4.* We compare the normalized training steps and wall-clock training time required to achieve comparable accuracy on the Qwen3-1.7B model. The results demonstrate that our D-ARL outperforms all asynchronous baselines in training efficiency.

**Experiment 1. Reasoning Performance Evaluation** In this subsection, we compare D-ARL with baselines in terms of reasoning performance. As shown in Table 1, D-ARL consistently outperforms all baselines across all benchmarks, demonstrating its superior effectiveness in improving reasoning performance. For Qwen3-1.7B, D-ARL achieves substantial performance gains over both synchronous and asynchronous baselines. Specifically, compared with the synchronous baseline GRPO, D-ARL improves the average reasoning performance by 12.2% across all benchmarks. When compared to the state-of-the-art (SOTA) asynchronous baseline CISPO, D-ARL yields a corresponding improvement of 6.4%. For the Qwen3-4B model, D-ARL achieves an average accuracy improvement of 10.6% over the synchronous baseline GRPO and 5.6% over the SOTA asynchronous baseline across all benchmarks. Overall, these results demonstrate the effectiveness of D-ARL in training performance. For more results, please refer to Appendix C.1.

**Experiment 2. Training Efficiency Evaluation** In this subsection, we compare D-ARL with the asynchronous baselines in terms of training efficiency, including both sample efficiency and training time. To ensure a fair comparison of sample efficiency, we set the target accuracy to the lowest value achieved among all baseline methods and measure the training steps required by each method to reach it. As shown in Figure 4a, D-ARL significantly outperforms the baselines in terms of sample efficiency for Qwen3-1.7B. Specifically, D-ARL achieves an average improvement of 34.7% over the SOTA baseline across all benchmarks. In terms of training time, Figure 4b shows that D-ARL reduces the wall-clock training time by an average of 37.2% compared with the SOTA baseline. Overall, these results demonstrate the superior training efficiency of our D-ARL, highlighting its advantages in training efficiency. Due to limited space, additional efficiency results for Qwen3-4B are provided in Appendix C.2.

*Table 2.* **Ablation study of D-ARL components on Qwen3-1.7B.** We evaluate the impact of Distribution-matched Sample Selection and Multi-Behavior Policy Optimization. Results show that removing either component leads to a significant performance drop, confirming their synergistic contribution to the overall effectiveness of D-ARL.

| Datasets | GSM8K | LightEval | MATH-500 | AIME24 | LiveCodeBench | HumanEval | **Average** |
|---|---|---|---|---|---|---|---|
| **Method** | Acc (%) ↑ | Acc (%) ↑ | Acc (%) ↑ | Acc (%) ↑ | Acc (%) ↑ | Acc (%) ↑ | Acc (%) ↑ |
| D-ARL (Ours) | **84.8** | **65.7** | **78.7** | **26.7** | **54.4** | **81.1** | **65.2** |
| D-ARL w/o Selection | 83.2 | 62.3 | 75.3 | 23.3 | 49.8 | 68.3 | 60.4 |
| D-ARL w/o Optimization | 69.1 | 48.7 | 75.7 | 20.0 | 46.9 | 72.0 | 55.4 |
| D-ARL w/o Selection and Optimization | 69.1 | 48.1 | 72.2 | 16.7 | 44.8 | 63.6 | 52.4 |

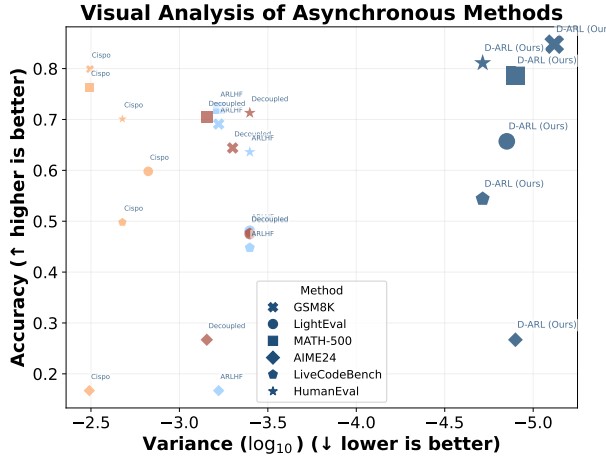

*(a)* Variance and Accuracy Analysis for Qwen3-1.7B

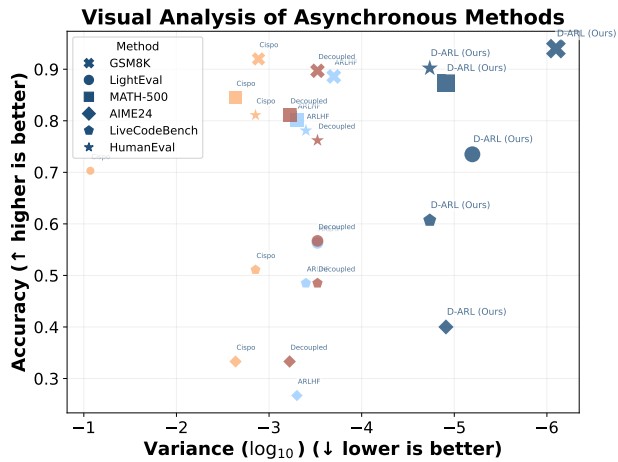

*(b)* Variance and Accuracy Analysis for Qwen3-4B

*Figure 5.* **Visual analysis of asynchronous data selected during training.** Scatter plots of sample variance (x-axis) versus accuracy (y-axis) across different datasets and methods for (a) Qwen3-1.7B and (b) Qwen3-4B. On each dataset, D-ARL consistently achieves lower variance and higher accuracy than standard asynchronous baselines.

**Experiment 3. Ablation Study** In this subsection, we conduct an ablation study to understand the individual contribution of each component in D-ARL. To this end, we compare our D-ARL with two representative variants: (i) D-ARL without distribution-matched sample selection and (ii) D-ARL without multi-behavior policy optimization. All variants are evaluated on six widely used open-source benchmarks under identical experimental settings. The results in Table 2 suggest the following two conclusions. First, D-ARL consistently outperforms the variant without distribution-matched sample selection in terms of reasoning performance. In particular, on the Qwen3-1.7B model, D-ARL achieves an average accuracy improvement of 4.8% across all benchmarks. This result highlights the critical role of distribution-matched asynchronous samples in stabilizing the training process and improving optimization effectiveness. Second, D-ARL also achieves substantially better performance than the variant without multi-behavior policy optimization. Specifically, D-ARL yields an average accuracy improvement of 9.8% across all benchmarks. This demonstrates that the proposed multi-behavior policy optimization mechanism is essential for effectively aggregating and exploiting information from heterogeneous asynchronous data sources during policy updates. Overall, the ablation results confirm that both components are indispensable to the effectiveness of D-ARL, and their combination is crucial for achieving efficient training. Due to limited space, we provide more ablation results in Appendix C.3.

**Experiment 4. Analysis of D-ARL** In this subsection, we provide a deep analysis to further illustrate the advantages of D-ARL from the perspective of asynchronous data selection. Specifically, we first visualize the sources of the selected samples to better understand how D-ARL exploits heterogeneous asynchronous data. As shown in Figure 8, the selected asynchronous samples are drawn from a wide range of candidate data sources, indicating that D-ARL is able to effectively leverage the diversity of asynchronously generated data and select high-potential samples, rather than relying on a narrow subset.

Moreover, we show that our sample selection mechanism yields more stable training by reducing the variance of the RL return estimator. Specifically, Figure 5 presents scatter plots of variance versus accuracy across different datasets and methods for Qwen3-1.7B and Qwen3-4B. For each dataset, D-ARL consistently achieves both lower sam-

*Table 3.* Percentage of total training time spent on sample selection across models of different sizes and mathematical benchmarks.

| Datasets | Mathematical | | | Average |
|---|---|---|---|---|
| | GSM8K | LightEval | AIME24 | |
| Models | Time Ratio (%) | Time Ratio (%) | Time Ratio (%) | Time Ratio (%) |
| Qwen3-1.7B | 12.6 | 13.3 | 15.1 | 13.7 |
| Qwen3-4B | 8.2 | 9.7 | 21.9 | 13.3 |

ple variance and higher accuracy compared to other asynchronous baselines, exhibiting a clear advantage. This indicates that D-ARL preferentially selects samples that are not only informative but also statistically more stable, which is critical for stabilizing asynchronous training.

Furthermore, we show that the sample selection step accounts for only a small fraction of the overall training time. As shown in Table 3, for Qwen3-1.7B, the time spent on sample selection constitutes only 13.7% of the total training time on three mathematical benchmarks, respectively. For the larger Qwen3-4B model, the corresponding proportions remain similarly low, at 13.3% on three mathematical benchmarks. These consistently small overheads indicate that our method introduces negligible additional computational cost and is readily applicable to large-scale RL training.

Overall, these results provide intuitive evidence that D-ARL can effectively utilize heterogeneous asynchronous data and identify high-quality samples that simultaneously improve training stability and performance. Due to limited space, we defer more results and analysis in Appendix C.4.

## 6. Related Work

**Asynchronous Sample Selection for Policy Optimization.**
A common strategy in existing asynchronous RL systems is to control data freshness through heuristic sample selection mechanisms. Most approaches maintain a bounded replay buffer that stores trajectories generated by the most recent behavior policy, and only samples from this buffer are used for policy updates. After one or a few optimization steps, the buffered data are discarded and replaced by newly collected samples. Such designs implicitly assume that samples generated by temporally adjacent policies are sufficiently aligned with the current policy, thereby reducing off-policy effects without explicit distribution correction. This strategy has been widely adopted in large-scale asynchronous RL systems to improve stability and system throughput (Fu et al., 2025; Lu et al., 2025). However, this heuristic freshness-based selection neither measures distribution mismatch nor exploits potential asynchronous data generated by older policies, leading to suboptimal training performance.

**Off-Policy Optimization Methods for Asynchronous RL.**
Asynchronous RL inevitably introduces off-policy data due to stale behavior policies, making direct importance sam-

pling (IS) correction prone to high variance when the discrepancy between the behavior policy $q$ and the target policy $p$ grows. To stabilize training under such a distribution mismatch, existing off-policy optimization methods for asynchronous RL can be broadly categorized into two lines. (1). *Gradient truncation methods* aim to control instability by explicitly restricting the influence of off-policy samples on policy updates. Representative approaches such as Decoupled PPO (Hilton et al., 2022) decouple data collection and policy optimization while truncating or clipping gradients to prevent large policy updates caused by highly off-policy data. These methods improve stability but may discard useful information from asynchronous samples. (2). *Important sampling optimization methods* focus on reducing the variance of IS-based corrections. Techniques such as CISPO (Chen et al., 2025), TOPR (Roux et al., 2025), Truncated IS (Espeholt et al., 2018) modify the importance ratios through truncation, reweighting, or normalization to balance bias and variance. While effective in practice, these methods typically apply uniform corrections to all off-policy data, without explicitly accounting for the degree of distribution alignment between the data and the current policy.

## 7. Conclusion

Asynchronous reinforcement learning offers significant potential for accelerating the post-training of large language models (LLMs), but distributional mismatches between stale behavior policies and the current policy lead to unstable training and performance degradation. To address this challenge, we proposed D-ARL, a distribution-matched asynchronous RL framework that selects high-quality samples aligned with the current policy and leverages them via multi-behavior policy optimization. Experiments across six widely used reasoning benchmarks demonstrate that D-ARL consistently outperforms state-of-the-art asynchronous methods, achieving an average improvement of 6.4% in reasoning performance and 34.7% in sample efficiency.

## Impact Statement

This work advances efficient and scalable reinforcement learning for large language models post-training. By addressing the distributional mismatch in asynchronous pipelines, D-ARL improves training stability, sample efficiency, and resource utilization. This reduces wasted computation and lowers the cost and environmental footprint of LLM training. Beyond empirical gains, our framework highlights the importance of principled data selection and policy-aware optimization when learning from heterogeneous asynchronous data sources. These insights provide a general perspective on stabilizing off-policy learning and may extend to other domains, including robotics, recommendation systems, and multi-agent learning.

## Acknowledgments

This work was supported in part by National Key R&D Program of China under contract 2022ZD0119801, National Nature Science Foundations of China grants U23A20388 and 62021001. This work was supported in part by Huawei as well. We would like to thank all the anonymous reviewers for their insightful comments.

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

# A. Theoretical Analysis

## A.1. Proof of Theorem 4.1

*Proof.* We aim to find the sampling distribution $\pi_q$ that minimizes the variance of the importance-sampling estimator

$$\hat{J}_{\pi_q} = \frac{1}{n} \sum_{i=1}^{n} \frac{\pi_p(\tau_i)}{\pi_q(\tau_i)} R(\tau_i),$$

where $\{\tau_i\}_{i=1}^{n}$ are i.i.d. samples drawn from $\pi_q$.

Since the samples are independent, the variance decomposes as

$$\mathrm{Var}[\hat{J}_{\pi_q}] = \frac{1}{n} \mathrm{Var}_{\tau \sim \pi_q} \left[ \frac{\pi_p(\tau)}{\pi_q(\tau)} R(\tau) \right].$$

Using the identity $\mathrm{Var}[X] = \mathbb{E}[X^2] - (\mathbb{E}[X])^2$ and noting that

$$\mathbb{E}_{\tau \sim \pi_q} \left[ \frac{\pi_p(\tau)}{\pi_q(\tau)} R(\tau) \right] = \mathbb{E}_{\tau \sim \pi_p}[R(\tau)] \triangleq J,$$

which is independent of $\pi_q$, we obtain

$$\mathrm{Var}[\hat{J}_{\pi_q}] = \frac{1}{n} \left( \mathbb{E}_{\tau \sim \pi_q} \left[ \frac{\pi_p^2(\tau) R^2(\tau)}{\pi_q^2(\tau)} \right] - J^2 \right).$$

Therefore, minimizing $\mathrm{Var}[\hat{J}_{\pi_q}]$ is equivalent to minimizing

$$\mathbb{E}_{\tau \sim \pi_q} \left[ \frac{\pi_p^2(\tau) R^2(\tau)}{\pi_q^2(\tau)} \right] = \int \frac{\pi_p^2(\tau) R^2(\tau)}{\pi_q(\tau)} \, d\tau,$$

subject to the normalization constraint $\int \pi_q(\tau) \, d\tau = 1$.

We introduce the Lagrangian

$$\mathcal{L}(\pi_q; \lambda) = \int \frac{\pi_p^2(\tau) R^2(\tau)}{\pi_q(\tau)} \, d\tau + \lambda \left( \int \pi_q(\tau) \, d\tau - 1 \right).$$

Taking the functional derivative of $\mathcal{L}$ with respect to $\pi_q(\tau)$ and setting it to zero yields

$$-\frac{\pi_p^2(\tau) R^2(\tau)}{\pi_q^2(\tau)} + \lambda = 0,$$

which gives the optimal solution

$$\pi_{q^*}(\tau) = \frac{\pi_p(\tau) \, |R(\tau)|}{\sqrt{\lambda}}.$$

Enforcing the normalization constraint,

$$\int \pi_{q^*}(\tau) \, d\tau = 1 \quad \Rightarrow \quad \sqrt{\lambda} = \mathbb{E}_{\tau \sim \pi_p}[|R(\tau)|].$$

In the common setting where $R(\tau) \geq 0$, this reduces to

$$\pi_{q^*}(\tau) = \frac{\pi_p(\tau) R(\tau)}{\mathbb{E}_{\tau \sim \pi_p}[R(\tau)]}. \qquad \square$$

## A.2. Proof of Theorem 4.3

*Proof.* We seek to minimize the variance of the estimator $\hat{J}_{\text{DARL}}$ with respect to the aggregation weights $\{w_i\}_{i=1}^N$. By definition, its variance can be written as

$$\text{Var}[\hat{J}_{\text{DARL}}] = \sum_{i=1}^N \frac{w_i^2}{n_i^*} \text{Var}_{\tau \sim \pi_{q_i}} \left[ \frac{\pi_p(\tau)}{\pi_{q_i}(\tau)} R(\tau) \right].$$

For notational simplicity, we denote

$$\sigma_i \triangleq \frac{1}{\text{Var}_{\tau \sim \pi_{q_i}} \left[ \frac{\pi_p(\tau)}{\pi_{q_i}(\tau)} R(\tau) \right]},$$

which yields

$$\text{Var}[\hat{J}_{\text{DARL}}] = \sum_{i=1}^N \frac{w_i^2}{n_i^* \sigma_i}.$$

We minimize this expression subject to the constraint $\sum_{i=1}^N w_i = 1$. Introducing a Lagrange multiplier $\lambda$, the Lagrangian is

$$\mathcal{L}(w_1, \dots, w_N; \lambda) = \sum_{i=1}^N \frac{w_i^2}{n_i^* \sigma_i} - \lambda \left( \sum_{i=1}^N w_i - 1 \right).$$

Taking the first-order partial derivative with respect to $w_i$ and setting it to zero gives

$$\frac{\partial \mathcal{L}}{\partial w_i} = \frac{2w_i}{n_i^* \sigma_i} - \lambda = 0,$$

which implies

$$w_i = \frac{\lambda}{2} n_i^* \sigma_i.$$

Substituting this expression into the constraint $\sum_{i=1}^N w_i = 1$, we obtain

$$\lambda = \frac{2}{\sum_{j=1}^N n_j^* \sigma_j}.$$

Therefore, the optimal weights are given by

$$w_i = \frac{n_i^* \sigma_i}{\sum_{j=1}^N n_j^* \sigma_j}. \qquad \square$$

# B. More Details on the Background and Related Work

**Synchronous RL Post-Training** Reinforcement learning (RL) post-training for large language models typically follows a synchronous pipeline consisting of three stages: rollout, reward assignment, and policy optimization (Schulman et al., 2017; Xiao et al., 2022; Bai et al., 2026a;b; 2025). During rollout, the agent interacts with the environment over multiple turns, generating trajectories composed of state–action sequences (Liang et al., 2025b;a; 2026; Wang et al., 2024; Wang et al.). A reward model then assigns scalar feedback to each trajectory, which is subsequently used to update the policy parameters during training. In synchronous RL post-training, rollouts and policy updates are tightly coupled, requiring strict synchronization of model parameters at every training iteration. This design introduces execution barriers between data collection and optimization, often leading to limited scalability and suboptimal hardware utilization.

To illustrate a representative synchronous optimization paradigm, we focus on **Group Relative Policy Optimization (GRPO)** (Shao et al., 2024), a recently proposed critic-free policy optimization method tailored for large language model post-training. GRPO eliminates the need for explicit value function estimation by constructing advantage signals through relative comparisons among multiple responses generated for the same prompt. Specifically, given a prompt $q$, GRPO samples a group of $G$ output sequences $\{\tau_i\}_{i=1}^G$ using the current policy, where each sequence is associated with a scalar reward $r_i$. The normalized advantage for the $t$-th token of the $i$-th sequence is defined as:

$$\hat{A}_{i,t} = \frac{r_i - \text{mean}(\{r_j\}_{j=1}^G)}{\text{std}(\{r_j\}_{j=1}^G)}.$$

By standardizing rewards within each group, GRPO provides informative and discriminative learning signals even under sparse or low-variance reward settings, effectively emphasizing intra-group relative ranking. The GRPO objective is formulated as:

$$\mathcal{J}_{\mathrm{GRPO}}(\theta) =$$

$$\mathbb{E}_{q\sim P(Q),\{\tau_i\}_{i=1}^G\sim\pi_{\theta_{\mathrm{old}}}(\cdot|q)}\left[\frac{1}{G}\sum_{i=1}^G\frac{1}{|\tau_i|}\sum_{t=1}^{|\tau_i|}\Big(\min\Big(r_{i,t}(\theta)\hat{A}_{i,t},\mathrm{clip}(r_{i,t}(\theta),1-\epsilon,1+\epsilon)\hat{A}_{i,t}\Big)-\beta D_{\mathrm{KL}}(\pi_\theta\|\pi_{\mathrm{ref}})\Big)\right]$$

where $r_{i,t}(\theta) = \frac{\pi_\theta(\tau_{i,t}|q,\tau_{i,<t})}{\pi_{\theta_{\mathrm{old}}}(\tau_{i,t}|q,\tau_{i,<t})}$, $\epsilon$ controls the clipping range, $\beta$ is the regularization coefficient, and $D_{\mathrm{KL}}(\pi_\theta\|\pi_{\mathrm{ref}})$ denotes the KL divergence between the current policy and a reference policy.

Overall, GRPO provides a stable and efficient synchronous RL optimization framework for language model post-training, serving as a strong foundation for the asynchronous and forgetting-aware extensions introduced in subsequent sections.

**Multiple Important Sampling** Multiple Important Sampling (MIS) aims to estimate the performance of a target policy $\pi_p$ given episodes collected with a set of $N$ behavior policies $\{\pi_{q_i}\}_{i=1}^N$. In MIS frameworks, we have a data set $\{\tau_{i,1},\tau_{i,2},\cdots,\tau_{i,n_i^*}\}$ of $n_i^*$ samples collected independently with $\pi_{q_i}$, $i=1,2,\cdots,N$. The resulting estimator is given by:

$$\hat{J}_{\pi_p/\pi_{q_{1:N}}}^w = \sum_{i=1}^N\frac{1}{n_i^*}\sum_{k=1}^{n_i^*}w_i(\tau_{i,k})\frac{\pi_p(\tau_{i,k})}{\pi_{q_i}(\tau_{i,k})}R(\tau_{i,k})$$

where $w_i(\tau)$ is a partition of the unity, i.e., a collection of weight functions for which $w_i(\tau)\geq 0$ for all $i=1,2,\cdots,N$ and $\sum_{i=1}^N w_i(\tau)=1$ for all $\tau$. Traditional MIS uses sample-level weights, which increases the computational complexity and neglects the integrity of the data sampled from the same behavior policy. In our frameworks, the weights are replaced by policy-level weights, which better aggregate the information of data sampled from the same behavior policy and reduce the computational complexity. Our estimator is given by:

$$\hat{J}_{DARL} = \sum_{i=1}^N\frac{w_i}{n_i^*}\sum_{k=1}^{n_i^*}\frac{\pi_p(\tau_{i,k})}{\pi_{q_i}(\tau_{i,k})}R(\tau_{i,k}).$$

where $w_i$ is the optimal policy-level weights defined in Theorem 4.3.

## C. Additional Results

### C.1. More Results for Reasoning Performance Comparative Evaluation

Figure 6 provides the full training trajectories for both Qwen3-1.7B (top row) and Qwen3-4B (bottom row) across all six benchmarks. The results demonstrate that D-ARL consistently outperforms all baseline methods in both convergence speed and final performance. Specifically, D-ARL exhibits significantly higher sample efficiency, reaching comparable accuracy levels to the baselines in a fraction of the training time, while ultimately achieving higher asymptotic accuracy given the same computational budget. For Qwen3-1.7b, D-ARL yields an average relative improvement of 12.2% across all benchmarks compared to the synchronous baseline GRPO within identical training steps. This includes gains of 11.8% on mathematical reasoning and 13.1% on code generation tasks. For Qwen3-4b, the performance advantage remains robust, with an average improvement of 10.6% over GRPO (9.3% for math and 13.2% for code benchmarks). These learning curves highlight that D-ARL effectively mitigates the performance degradation typically associated with policy staleness in asynchronous RL, maintaining a training trajectory that is both more stable and more efficient than existing SOTA methods.

### C.2. More Results for Training Efficiency Comparative Evaluation

Figure 7 illustrates the superior training efficiency of D-ARL compared to baseline methods. Our approach achieves state-of-the-art performance with significantly reduced computational overhead and wall-clock time. In terms of training steps required to reach a target accuracy level, D-ARL demonstrates substantial gains over existing asynchronous techniques. On the Qwen3-4B model, D-ARL achieves an average sample efficiency improvement of 56.6% over Decoupled GRPO and 36.1% over CISPO across all benchmarks. For mathematical reasoning tasks specifically, these improvements stand

## Evaluation of Training Accuracy

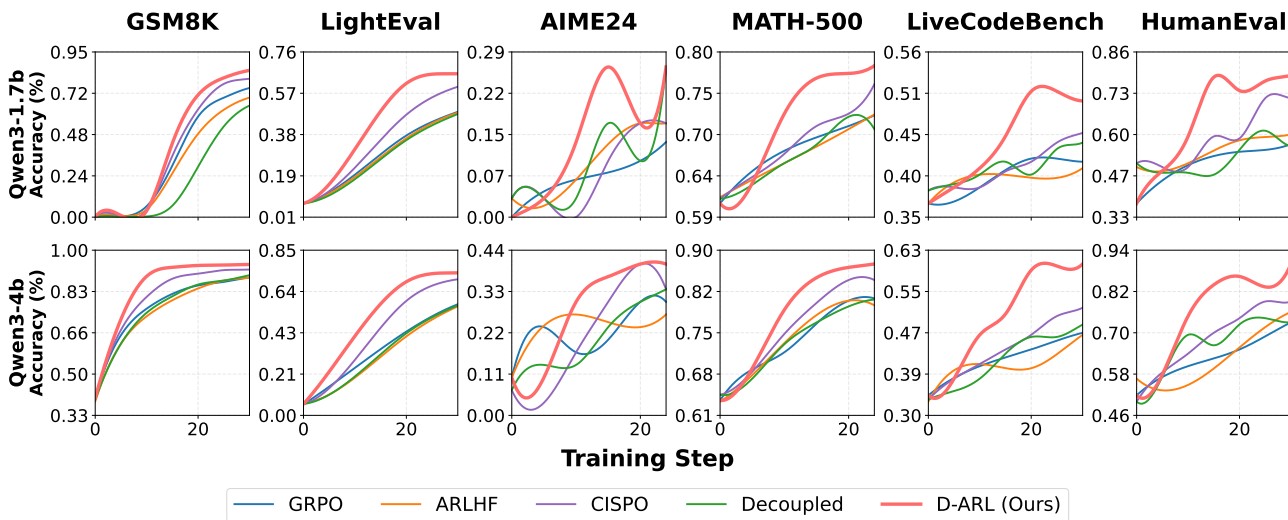

*Figure 6.* **Training curves on mathematical reasoning and code generation benchmarks.** We plot the evaluation accuracy against training steps for the Qwen3-1.7B (top) and Qwen3-4B (bottom) models. The results show that **D-ARL** (ours) consistently achieves faster convergence and higher final accuracy compared to both synchronous and asynchronous baselines.

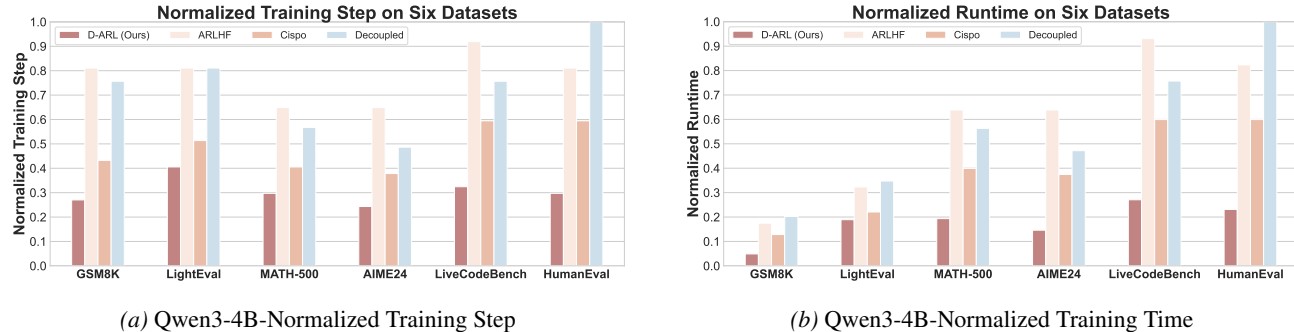

*(a)* Qwen3-4B-Normalized Training Step          *(b)* Qwen3-4B-Normalized Training Time

*Figure 7.* **Training efficiency comparison on Qwen3-4B across six benchmarks.** (a) Normalized training steps required to reach comparable performance, demonstrating the superior sample efficiency of **D-ARL**. (b) Normalized wall-clock training time, highlighting that **D-ARL** achieves the target accuracy with the lowest computational cost among all evaluated asynchronous methods.

at 53.0% and 30.2%, respectively. The gains are even more pronounced in code generation benchmarks, where D-ARL outperforms Decoupled GRPO and CISPO by 63.7% and 47.7% in sample efficiency. As shown in Figure 7b, D-ARL achieves the highest training throughput among all asynchronous candidates. Specifically, on mathematical benchmarks, D-ARL consumes 47.2% less runtime compared to the CISPO baseline while achieving comparable accuracy. For code generation tasks, the runtime reduction is even more significant at 58.1%. These results indicate that by effectively utilizing off-policy samples with lower policy divergence, D-ARL not only stabilizes the learning but also accelerates the training.

### C.3. More Ablation Study Results

In this subsection, we present additional ablation results on the Qwen3-4B model to further examine the contribution of each component in D-ARL. Following the same experimental protocol as in the main text, we compare D-ARL with two variants: (i) D-ARL without distribution-matched sample selection, and (ii) D-ARL without multi-behavior policy optimization. All methods are evaluated on the same six open-source benchmarks under identical settings. The results reported in Table 4 exhibit trends that are consistent with those observed for Qwen3-1.7B. First, D-ARL consistently outperforms the variant without distribution-matched sample selection in terms of reasoning performance. On Qwen3-4B, D-ARL achieves a notable average accuracy improvement of 3.8% across both mathematical and coding benchmarks, demonstrating that distribution-matched asynchronous samples remain important for stabilizing training and improving optimization effectiveness at larger model scales. Second, removing the multi-behavior policy optimization component

*Table 4.* **Ablation study of D-ARL components on Qwen3-4B.** We evaluate the impact of Distribution-Matched Sample Selection (DMSS) and Multi-Behavior Policy Optimization (MBPO). Results show that removing either component leads to a significant performance drop, confirming their synergistic contribution to the overall effectiveness of D-ARL.

| Datasets | GSM8K | LightEval | MATH-500 | AIME24 | LiveCodeBench | HumanEval | **Average** |
|---|---|---|---|---|---|---|---|
| Method | Acc (%) ↑ | Acc (%) ↑ | Acc (%) ↑ | Acc (%) ↑ | Acc (%) ↑ | Acc (%) ↑ | Acc (%) ↑ |
| D-ARL (Ours) | **94.0** | **73.5** | **87.3** | **40.0** | **60.7** | **90.2** | **74.3** |
| D-ARL w/o Selection | 92.5 | 72.3 | 85.5 | 36.7 | 54.8 | 81.1 | 70.5 |
| D-ARL w/o Optimization | 92.0 | 54.6 | 81.3 | 33.3 | 54.0 | 78.1 | 65.6 |
| D-ARL w/o Selection and Optimization | 88.6 | 56.3 | 80.1 | 26.7 | 48.5 | 78.1 | 63.1 |

*Figure 8.* **Visualization of multi-behavior policy sampling distributions.** The figure illustrates the relative sampling proportions of asynchronous training data generated by the last four behavior policies, denoted as $\pi_{q^*_{t-1}}, \pi_{q^*_{t-2}}, \pi_{q^*_{t-3}}$, and $\pi_{q^*_{t-4}}$. The area of each region approximately reflects the fraction of samples drawn from the corresponding behavior policy. Results are reported across different benchmarks and model scales, showing how training samples are collected from multiple historical policies during optimization.

leads to a more pronounced performance degradation. D-ARL yields a substantial reasoning performance improvement of 8.7% than the corresponding variant across all benchmarks, indicating that multi-behavior policy optimization is crucial for effectively aggregating and leveraging heterogeneous asynchronous data when training larger models. Overall, these results further confirm that both components of D-ARL are essential to its effectiveness and that their benefits consistently extend from smaller to larger model scales.

## C.4. More Visual Analysis Results

We provide additional visual analysis results on the multi-behavior policy distribution in Figure 8. Specifically, we visualize the contribution of the last four behavior policies, denoted as $\pi_{q^*_{t-1}}, \pi_{q^*_{t-2}}, \pi_{q^*_{t-3}}, \pi_{q^*_{t-4}}$, where $\pi_{q^*_{t-1}}$ corresponds to the most recent behavior policy and is therefore expected to be the closest to the current target policy. In the Figure 8, the area of each region approximately reflects the proportion of samples drawn from the corresponding behavior policy during off-policy training. A larger area indicates a higher sampling ratio. Note that among the six benchmarks considered, AIME24 and MATH-500 share the same training set, and thus exhibit identical policy distributions. A similar situation holds for LiveCodeBench and HumanEval. As shown in Figure 8, the commonly adopted assumption in asynchronous training—that more recent behavior policies are necessarily better aligned with the current policy—does not always hold in practice. We observe that samples generated by earlier behavior policies can also exhibit a high degree of alignment with the current policy, and in some cases contribute a dominant portion of the high-quality training data.

This phenomenon is particularly evident on the GSM8K benchmark. For Qwen3-1.7B and Qwen3-4B, samples collected from the most recent behavior policy $\pi_{q^*_{t-1}}$ account for only approximately 18.1% and 8.1% of the high-quality training samples, respectively. Remarkably, for Qwen3-4B, the behavior policy $\pi_{q^*_{t-3}}$ contributes as much as 71.0% of the high-

quality samples. Combined with the quantitative results reported in Figure 5 and the ablation studies in Appendix C.3, which demonstrate that higher-quality samples lead to substantially improved training outcomes, these observations highlight the effectiveness of our method in identifying distribution-matched and high-value samples from heterogeneous behavior policies. Overall, the results provide further evidence that D-ARL is capable of leveraging not only the most recent behavior policy, but also earlier policies, to efficiently exploit asynchronous data and enhance training efficiency.

## D. Details of Dataset Used in This Paper

In our experiments, we use a range of mathematical and code generation datasets to evaluate the reasoning capabilities of our methods.

### D.1. Mathematics-Related Datasets

- **GSM8K** (Cobbe et al., 2021) is a benchmark dataset for elementary school mathematics word problems, consisting of 8,500 high-quality problems aligned with K-12 curricula. Each problem is paired with a step-by-step solution, enabling evaluation of multi-step arithmetic and algebraic reasoning.

- **LightEval** (Hendrycks et al., 2021) is a lightweight, multi-task evaluation benchmark developed by Hugging Face, integrating mathematical reasoning, natural language understanding, and knowledge retrieval tasks. Its mathematical subset covers arithmetic, geometry, and logical deduction with concise, diverse problems, making it suitable for fast, low-overhead evaluation of general mathematical competence.

- **MATH-500** (Lightman et al., 2023) is a curated dataset of 500 middle- and high school-level problems, spanning algebra, geometry, trigonometry, probability, and statistics. Each problem is annotated with difficulty levels, enabling gradient-based evaluation of advanced mathematical reasoning beyond elementary benchmarks.

- **AIME24** (Zhang & Math-AI, 2024) is a subset of 30 competition-level problems from the 2024 AIME, designed for high-ability students. These problems require integrating multiple mathematical concepts and non-trivial reasoning steps, posing significant challenges to large language models.

### D.2. Code-Related Datasets

- **Livecodebench** (Jain et al., 2024) is a state-of-the-art benchmark for code generation and understanding, built on real-world GitHub code across 20+ programming languages (Python, Java, C++, JavaScript, etc.). It covers tasks including code completion, translation, bug fixing, and function implementation, with evaluation metrics for correctness, readability, and efficiency.

- **HumanEval** (Chen et al., 2021) is a widely used benchmark for functional correctness of code generated by language models. It consists of hand-crafted Python programming problems, each paired with a unit test suite to automatically verify solution correctness. HumanEval focuses on evaluating code generation accuracy, reasoning about function specifications, and handling edge cases.

### D.3. Training Data and Splits

Table 5 summarizes the sizes of the training and test sets for each benchmark. For datasets GSM8K and LightEval, we employ the official training splits. For the remaining benchmarks—AIME24, MATH-500, Livecodebench, and HumanEval—we utilize the **Guru-RL-92K** dataset (Cheng et al., 2025). Guru-RL-92K is a carefully curated RL benchmark, containing approximately 92K high-quality samples processed through a multi-stage curation pipeline to ensure domain diversity. Specifically, we select around 6.2K mathematical samples for AIME24 and MATH-500, and approximately 10K code samples for Livecodebench and HumanEval.

*Table 5.* The sizes of the training and test sets for each dataset.

| Benchmark | Training Set | Test Set |
|---|---|---|
| GSM8K | 7.5K | 1.5K |
| LIGHTEVAL | 7.5K | 5K |
| MATH-500 | 6.2K | 0.5K |
| AIME24 | 6.2K | 30 |
| LiveCodeBench | 10K | 500 |
| HumanEval | 10K | 164 |

# E. Details of Methods and Experimental Settings

## E.1. Details of Experimental Setup

In this section, we provide comprehensive details about our model training, baselines, and the hyperparameter setup.

**Implementation of the Training Details** For all experiments, models are trained for a single epoch due to computational budget constraints. Nevertheless, our empirical observations indicate that increasing the number of training epochs yields consistent performance trajectories. The random seed is fixed to 42 for all experiments to ensure reproducibility. During the rollout phase, the sampling temperature is set to 1.0 to maintain stable generation behavior. The maximum response length is tailored to the specific demands of each dataset: we set it to 1024 tokens for GSM8K and LightEval, while for more complex reasoning tasks such as MATH-500, AIME24, LiveCodeBench, and HumanEval, the maximum response length is increased to 4096 tokens. To ensure consistency between training and evaluation, the maximum response length is kept identical during both the rollout and evaluation phases. All experiments are conducted on a single node equipped with four NVIDIA A100 or A800 GPUs. Among them, two GPUs are allocated for rollout generation, while the remaining two are used for model training.

**Implementation Details of the Baselines** All baseline methods are implemented within a unified training framework to ensure fair comparison. Unless otherwise stated, all baselines share the same model architecture, rollout procedure, reward function, batch size, and optimizer settings. The only differences lie in the policy optimization objectives and the handling of off-policy data.

**GRPO** (Shao et al., 2024) The standard GRPO baseline follows the on-policy formulation, where multiple responses are sampled per prompt and group-wise reward normalization is used to compute relative advantages . We report two GRPO variants: a fully synchronous version (**GRPO**) and a one-step asynchronous variant (**ARLHF**). GRPO serves as a strong baseline under synchronous or near-synchronous settings, with the clipping parameter set to $\epsilon = 0.2$.

**ARLHF** (Noukhovitch et al., 2025a) ARLHF performs policy updates using samples generated by the policy from the previous training step, thereby introducing mild policy staleness. No explicit importance sampling correction is applied, which may lead to degraded performance as the distribution mismatch increases. The clipping parameter is set to $\epsilon = 0.2$.

**Decoupled** (Hilton et al., 2022) Decoupled GRPO introduces an auxiliary proximal policy to stabilize optimization under asynchronous training by factorizing the importance sampling ratio into short-term and long-term components. This design mitigates variance induced by stale samples and improves training stability. In our implementation, the proximal policy is instantiated as the current training policy, with $\epsilon = 0.2$.

**CISPO** (Chen et al., 2025) CISPO constrains the importance sampling ratio within an asymmetric interval around 1, enabling finer control over off-policy updates. For controlled comparison, we use symmetric bounds $\epsilon_{\text{low}}^{\text{IS}} = \epsilon_{\text{high}}^{\text{IS}} = 0.2$, such that the primary difference from GRPO stems from the objective formulation.

For all baselines, we use the same number of rollout samples and training steps, without introducing additional heuristics or task-specific modifications.

**Hyperparameters** Table 6 summarizes the key hyperparameters used across all experiments, including model configuration, rollout settings, and optimization-related parameters.

## E.2. More Details about the Distribution-matched sample selection mechanism

### E.2.1. More details about the Minimum-Variance Behavior Policy

The Minimum-Variance behavior policy $\pi_{q^*}$ defined in Theorem 4.1 is given by

$$\pi_{q^*}(\tau) = \frac{\pi_p(\tau)\,|R(\tau)|}{\mathbb{E}_{\tau \sim \pi_p}[R(\tau)]}.$$

In asynchronous reinforcement learning, at time step $T$, the trajectories $\tau$ generated by the current policy $\pi_p^{(T)}$ are not yet available for computing the exact expectation in the denominator. However, all historical trajectories collected in previous time steps $t = 1, \ldots, T-1$ are known. Let $\mathcal{T}_t$ denote the set of trajectories collected at time step $t$. We first compute the

*Table 6.* We provide comprehensive implementation details, including the configurations.

| Parameter | Value |
|---|---|
| Model Kwargs | |
| Max prompt length | 512 |
| Max response length for GSM8K and LightEval | 1024 |
| Max response length for MATH-500 and AIME24 | 4096 |
| Max response length for LiveCodeBench and HumanEval | 4096 |
| Train Kwargs | |
| Replay Buffer Length | 4 |
| Train data batch size | 256 |
| Optimizer learning rate | 1e-6 |
| KL loss coefficient | 0.001 |
| Clipping threshold | 0.2 |
| Clipping threshold c | 3.0 |
| Rollout.n | 4 |
| Sample warmup | 5 |

average return within each time step:

$$\bar{R}_t = \frac{1}{|\mathcal{T}_t|} \sum_{\tau \in \mathcal{T}_t} R(\tau),$$

and then approximate the expected return under the current policy by averaging over all past time steps:

$$\mathbb{E}_{\tau \sim \pi_p^{(T)}}[R(\tau)] \approx \frac{1}{T-1} \sum_{t=1}^{T-1} \bar{R}_t.$$

Substituting this approximation into the optimal behavior policy yields the operational form:

$$\hat{\pi}_{q^*}^{(T)}(\tau) = \frac{\pi_p^{(T)}(\tau)\,|R(\tau)|}{\frac{1}{T-1} \sum_{t=1}^{T-1} \bar{R}_t}.$$

This two-level averaging procedure leverages historical trajectory data to estimate the current expected return, enabling the selection of trajectories that are both aligned with the current policy and have high absolute rewards, thereby minimizing the variance of the importance-sampled return estimator $\hat{J}_{\pi_q}$.

### E.2.2. MORE DETAILS ABOUT THE OFF-POLICY DEGREE METRIC

The off-policy degree metric $\deg(\tau)$ defined in Definition 4.2 is designed to quantify the alignment of a trajectory $\tau$ with the optimal behavior policy $\pi_{q^*}$. Intuitively, if the probability of $\tau$ under the behavior policy $\pi_q$ is close to that of the optimal policy, then $\tau$ can be considered nearly on-policy. Formally, for a trajectory $\tau$ sampled from $\pi_q$, the off-policy degree is

$$\deg(\tau) = \frac{|\pi_{q^*}(\tau) - \pi_q(\tau)|}{\max\{\pi_{q^*}(\tau), \pi_q(\tau)\}}.$$

This design has the following properties:

1. **Normalized discrepancy:** By dividing the absolute difference by the maximum of $\pi_{q^*}(\tau)$ and $\pi_q(\tau)$, the metric is normalized to the range $[0, 1]$, allowing meaningful comparison across samples with different probabilities.

2. **Low off-policy degree indicates alignment:** When $\pi_q(\tau) \approx \pi_{q^*}(\tau)$, the numerator is small and thus $\deg(\tau)$ is close to 0, reflecting a low off-policy degree.

3. **Sensitive to large deviations:** If $\pi_q(\tau)$ differs substantially from $\pi_{q^*}(\tau)$, $\deg(\tau)$ approaches 1, signaling that the sample may induce high variance if used for policy updates.

Using this metric, we compute $\deg(\tau)$ for all samples in the replay buffer and select the top $k$ samples with the lowest off-policy degrees for asynchronous policy optimization. In this way, the selected samples are both closely aligned with $\pi_{q^*}$ and contribute to stable and efficient training.

### E.3. More Details about the Multi-behavior Policy Optimization method

In practice, when implementing the multi-behavior policy optimization for D-ARL, we make two modifications to the return estimator to ensure stable and correct gradient computation. Specifically, for each behavior policy $\pi_{q_i}$ and its associated trajectories $\{\tau_{i,k}\}$, the contribution to the return estimator is computed as

$$\hat{J}_{\text{DARL}} = \sum_{i=1}^{N} w_i \left( \frac{1}{n_i^*} \sum_{k=1}^{n_i^*} \frac{\pi_p(\tau_{i,k})}{\pi_{q_i}(\tau_{i,k})} R(\tau_{i,k}) \right).$$

First, to prevent the importance sampling (IS) ratio from introducing undesired gradients, we apply a stop-gradient operation (sg) to the IS term (Chen et al., 2025; Espeholt et al., 2018). This ensures that the ratio only reweights the trajectories for variance correction without affecting the gradient computation with respect to the current policy parameters $\theta$. Second, we multiply the reward $R(\tau_{i,k})$ by $\log \pi_\theta(\tau_{i,k})$ when performing backpropagation. This modification preserves the correct gradient signal for policy optimization while keeping the reweighting effect of the IS ratio independent of the gradient flow. With these modifications, the final operational return estimator used for multi-behavior policy optimization is

$$\hat{J}_{\text{DARL}} = \sum_{i=1}^{N} w_i \left( \frac{1}{n_i^*} \sum_{k=1}^{n_i^*} \text{sg}\!\left[ \frac{\pi_p(\tau_{i,k})}{\pi_{q_i}(\tau_{i,k})} \right] R(\tau_{i,k}) \log \pi_\theta(\tau_{i,k}) \right).$$

Consequently, the combined design ensures that: (1). The importance sampling ratios properly correct for distribution mismatch across multiple behavior policies without introducing biased gradients, and (2). The policy gradient is computed solely with respect to $\log \pi_\theta(\tau)$, allowing stable and unbiased updates of the target policy.

---

**Algorithm 1** Distribution-Matched Asynchronous Reinforcement Learning (D-ARL)

---

**Require:** Initial policy $\pi_p$, replay buffer $\mathcal{B}$ containing samples from last $K$ behavior policies $\{\pi_{q_i}\}_{i=1}^{K}$, selection size $k$, training steps $T$
**Ensure:** Updated policy parameters $\theta$
  Initialize policy parameters $\theta$                                        ▷ Policy Initialization
  **for** training step $t = 1$ to $T$ **do**
    **for** each behavior policy $\pi_{q_i}$ **do**
      Collect trajectories $\{\tau_{i,j}\}$ and store in $\mathcal{B}$
    **end for**
    **for** each trajectory $\tau \in \mathcal{B}$ **do**
      Estimate $\pi_{q^*}(\tau)$                ▷ Minimum-Variance Behavior Distribution
      Compute off-policy degree $\deg(\tau)$
    **end for**
    Select top-$k$ trajectories to form $\mathcal{B}^*$        ▷ Distribution-matched Sample Selection
    **for** each behavior policy $\pi_{q_i}$ **do**
      Compute $n_i^*$ and variance $\sigma_i$
      Compute weight $w_i$                    ▷ Optimal Policy-level Weight
    **end for**
    Compute aggregated return $\hat{J}_{\text{DARL}}$       ▷ Multi-behavior Policy Optimization
    Update $\theta$ via policy gradient
  **end for**

---

