# OpenReview forum: "D-ARL: A Distribution-Matched Asynchronous Reinforcement Learning Framework for Language Reasoning"
_ICML.cc/2026/Conference — ICML 2026 regular_

### Official Review · Reviewer_9vbQ · 2026-03-04

**Soundness:** 3
**Presentation:** 3
**Significance:** 3
**Originality:** 3
**Overall Recommendation:** 5
**Confidence:** 4

**Summary:**

In this paper, the authors aim to address the instability of asynchronous reinforcement learning. They propose D-ARL, which maintains a replay buffer, and introduce a variance-guided metric to select distribution-matched samples for RL training. The experimental results demonstrate the strong effectiveness of the proposed method. Overall, I think this is a solid paper.

**Compliance With Llm Reviewing Policy:**

Affirmed.

**Final Justification:**

The authors have successfully addressed my concerns during the rebuttal period. Thereby I raise my score from 4 to 5.

**Key Questions For Authors:**

1. Could the authors explain Figure 2b in more detail? In particular, why is the probability of $\pi_{q^*_{t-3}}$ already close to 0.95 at training step 5? Also, regarding Lines 149–152 (“only about 18.1% of these samples originate from the most recent behavior policy for Qwen3-1.7B on the GSM8K benchmark, indicating that historical behavior policies can also produce high-quality samples”), could the authors clarify precisely what it means?

2. The experimental results are very impressive: the proposed method achieves substantially better performance than the baselines while using much less compute. Do the authors plan to open-source the code to facilitate reproduction of the results reported in the paper?

**Limitations:**

It seems that the authors do not discuss the limitations.

**Strengths And Weaknesses:**

### Strength

1. The paper is well organized and easy to follow. The motivation is clearly stated, and the presentation stays focused on a single, coherent line of argument.

2. The proposed method is somewhat complex, but most of the new design choices are supported by clear theoretical guarantees.

3. The experimental results are genuinely surprising: the method substantially outperforms the baseline algorithms while requiring less computational cost.

### Weakness

I do not see any major weaknesses in the paper; my concerns are mostly minor. Some figures would benefit from more detailed explanations. In addition, the notation is overly complicated, with too many subscripts. Simplifying the notation would improve readability.

---

> ### Author Rebuttal · Authors · 2026-03-31
>
> Dear Reviewer 9vbQ,
>
> We greatly appreciate your careful reading and constructive feedback! We have provided further responses as follows. We sincerely hope that our responses could properly address all your concerns. If so, we would deeply appreciate it if you could raise your score. If not, please let us know your further concerns, and we will continue actively responding to your comments and improving our submission.
>
> ### Weakness
> > **I do not see any major weaknesses in the paper; my concerns are mostly minor. Some figures would benefit from more detailed explanations. In addition, the notation is overly complicated, with too many subscripts. Simplifying the notation would improve readability.**
>
> We sincerely appreciate the reviewer for the positive feedback and helpful suggestions. Following this comment, **we will improve the clarity of several figures by providing more detailed explanations in their captions and the main text**. In addition, we will revise the notation to simplify expressions and reduce the number of subscripts where possible, in order to improve overall readability.
>
> ### Question 1.1
> > **Explain Figure 2b in more detail: Why is the probability of $\pi_{q^*_{t-3}}$ already close to 0.95 at training step 5?**
>
> We thank the reviewer for this insightful question. The high probability of $\pi_{q^*_{t-3}}$ at an early training stage (e.g., step 5) can be attributed to the relatively small policy updates between consecutive steps. **In the early phase of training, neighboring policies (such as $\pi_{t-3}$) are still highly similar to the current policy, resulting in a high likelihood under the optimal behavior policy $\pi_{q^*}$**. Moreover, since the off-policy degree is determined by reward-weighted likelihood, trajectories generated by slightly earlier policies can still receive high probabilities if their rewards remain competitive. This effect is particularly pronounced in the early training stage, where the policy has not diverged significantly.
>
> Further, we would like to emphasize that **this is not a universal phenomenon**. The degree of distributional similarity between policies can **vary significantly depending on the model architecture, training dataset, and optimization dynamics**. As a result, the exact probability values may differ across settings.
>
> ### Question 1.2
> > **Regarding Lines 149–152 (“only about 18.1% of these samples originate from the most recent behavior policy for Qwen3-1.7B on the GSM8K benchmark, indicating that historical behavior policies can also produce high-quality samples”), could the authors clarify precisely what it means?**
>
> We sincerely appreciate the reviewer for this question and the opportunity to clarify. Here, **the “most recent behavior policy” refers to the policy at step $t-1$**, i.e., the latest policy available for data collection when performing the update at step $t$.
>
> The reported 18.1% indicates that, **among the selected top-quality samples (based on our selection mechanism), only 18.1% are generated by this most recent behavior policy ($t-1$)**, while the remaining samples come from trajectories produced by earlier (historical) policies stored in the replay buffer. This observation suggests that **high-quality samples are not exclusively produced by the most recent policy,** whereas traditional methods typically rely solely on samples generated by the latest policy. Instead, trajectories generated by earlier policies can still remain valuable and competitive in terms of reward, and thus are frequently selected during training. This supports our motivation for leveraging replay-buffer data, as it demonstrates that historical policies can provide meaningful and complementary training signals.
>
> ### Question 2
> > **Do the authors plan to open-source the code to facilitate reproduction of the results reported in the paper?**
>
> We sincerely thank the reviewer for this helpful question. **We plan to open-source our code to facilitate reproducibility and support future research**, and will include the release details in the final version.
>
> ### Limitations
> > **It seems that the authors do not discuss the limitations.**
>
> We sincerely thank the reviewer for pointing out the weaknesses of our paper. In the revised version, we will include a dedicated Limitations section, where we explicitly discuss the weaknesses and concerns raised by the reviewers, including the points mentioned above.

---

> > ### Author Rebuttal · Reviewer_9vbQ · 2026-04-03
> >
> > I thank the authors for their detailed explanations. As my concerns have been largely resolved, I will raise my score and confidence level accordingly.

---

> > > ### Author Response · Authors · 2026-04-03
> > >
> > > Dear Reviewer 9vbQ,
> > >
> > > Thank you very much for your time and for reading our rebuttal! We sincerely appreciate your positive feedback and your continued support！
> > >
> > > Best regards,
> > >
> > > Authors of Submission 20518

---

### Official Review · Reviewer_f3gZ · 2026-03-12

**Soundness:** 2
**Presentation:** 3
**Significance:** 3
**Originality:** 3
**Overall Recommendation:** 3
**Confidence:** 4

**Summary:**

This paper studies asynchronous RL for LLM post-training and focuses on the mismatch between stale behavior policies and the current policy in asynchronous pipelines. The proposed method, D-ARL, maintains a replay buffer containing trajectories generated by recent behavior policies and selects samples using an off-policy degree metric motivated by a minimum-variance behavior-policy analysis. The selected trajectories are then used in a multi-behavior policy optimization procedure. Experiments on six reasoning and coding benchmarks using Qwen3-1.7B and Qwen3-4B report an average 6.4% reasoning improvement over the strongest asynchronous baseline and a 34.7% gain in sample efficiency.

**Compliance With Llm Reviewing Policy:**

Affirmed.

**Key Questions For Authors:**

1. Can the authors more explicitly reconcile Theorem 4.1 with the implemented off-policy degree? In particular, what exact quantity is computed in practice, and how should readers interpret it relative to the theorem? A clear answer here could improve my soundness assessment.
2. Can the authors report multi-seed results for the main benchmark table, even if only on a subset of the datasets? If the gains remain stable, this would materially strengthen my confidence in the empirical results.
3. How sensitive are the conclusions to the CISPO configuration, given that the implementation uses symmetric clipping bounds? If the results remain strong under a more faithful CISPO setup, that would improve my view of the comparison.
4. What is the computational overhead of the replay-buffer selection step relative to the underlying GRPO update? A clearer answer would help me assess the practical significance of the method.
5. Do the authors have any evidence that the method transfers beyond the current Qwen3 plus GRPO-style setting? Even limited additional evidence would strengthen the paper's generality claims.

**Limitations:**

No. The paper should more directly discuss the single-seed evaluation, the gap between the theorem and the implemented heuristic, the dependence on baseline configurations, and the limited empirical scope relative to the framing.

**Strengths And Weaknesses:**

Strengths:
- The paper addresses an important practical problem in asynchronous RL for LLM training. Distribution mismatch from stale policies is a real bottleneck in decoupled rollout-and-update pipelines, so the problem choice is well motivated.
- The method is conceptually clear. The framework has two understandable parts: selecting better asynchronous data from the buffer and then optimizing over multi-source samples.
- The empirical study is reasonably broad for a first paper in this space: two model sizes, six reasoning/coding benchmarks, and both performance and efficiency metrics.

Weaknesses:
- My main concern is the connection between the theoretical derivation and the implemented algorithm. Theorem 4.1 is stated in terms of an importance-sampling estimator with an abstract return term, while the actual method uses an off-policy degree computed from approximations based on historical trajectories. The paper does not make fully clear how closely the implemented score tracks the quantity studied in the theorem.
- Robustness evidence is limited. Appendix E.1 states that all experiments use a fixed random seed of 42 and train for one epoch. For RL-style post-training, the absence of multi-seed reporting makes it harder to know whether the headline gains are stable.
- Baseline comparability deserves more discussion. The appendix states that CISPO is run with symmetric clipping bounds for controlled comparison, even though the method is introduced as asymmetric, and Decoupled is adapted inside the unified framework. These choices may be reasonable, but the paper should discuss more explicitly how much they may affect the comparison.
- The framing is somewhat broader than the evidence. All experiments are within a unified VERL-based setup, on Qwen3 models, with a GRPO-style training pipeline. This is promising evidence, but it is not yet broad evidence of generality across objectives, model families, or training infrastructures.
- The paper does not include a dedicated limitations section addressing these issues directly.

---

> ### Author Rebuttal · Authors · 2026-03-31
>
> Dear Reviewer f3gZ,
>
> We greatly appreciate your careful reading and constructive feedback! We have provided further responses as follows. We sincerely hope that our responses could properly address all your concerns. If so, we would deeply appreciate it if you could raise your score. If not, please let us know your further concerns, and we will continue actively responding to your comments and improving our submission. Moreover, due to ICML's limited response length for rebuttal, **we provide additional complementary results in the anonymous link.**
>
> ### Weakness 1 & Question 1
> > **Can the authors more explicitly reconcile Theorem 4.1 with the implemented off-policy degree?**
>
> We sincerely appreciate the reviewer for the opportunity to clarify our algorithm. In Theorem 4.1, we derive the optimal behavior policy $\pi_q^*(\tau) = \frac{\pi_p(\tau)R(\tau)}{E_{\tau \sim \pi_p}[R(\tau)]}$, which is used to define the off-policy degree in practice. **However, the term $E_{\tau \sim \pi_p}[R(\tau)]$ cannot be computed exactly**, as it requires sampling from the current policy $\pi_p$. In the asynchronous setting, **$\pi_p$ corresponds to the policy at step $t$, while only trajectories from previous policies (steps $0$ to $t-1$) are available**, making direct estimation infeasible.
>
> To address this, we approximate $E_{\tau \sim \pi_p}[R(\tau)]$ using historical trajectories. Empirically, this expectation remains relatively stable throughout training, so we use the **mean reward over all historical trajectories as an estimate**. As shown in Table a (see ***https://i.ibb.co/DPMGXmJ9/Theorem-explanation.png***), the relative difference—defined as $\frac{|\text{Est} - \text{True}|}{|\text{True}|}$—is **14.3% and 15.9%** for Qwen3-1.7B and Qwen3-4B, respectively. This suggests that our approximation **strikes a reasonable balance between accuracy and simplicity**, while leaving room for future improvements.
>
> ### Weakness 2 & Question 2
> > **Report multi-seed results for the main benchmark table.**
>
> We sincerely appreciate the reviewer for this helpful suggestion. Following this comment, we conduct multi-seed experiments on Qwen3-4B across four reasoning benchmarks.
>
> The results in Table b (see ***https://i.ibb.co/Rkx61HSV/multi-seed.png***) show that **our method achieves consistently low variance across three seeds (only 3.35% of the mean)**, indicating strong robustness and stability. We will include these results in the revised version.
>
> ### Weakness 3 & Question 3
> > **How sensitive are the conclusions to the CISPO configuration?**
>
> To evaluate sensitivity to the CISPO configuration, we conduct ablations with both **asymmetric** $(\epsilon_{\text{high}}, \epsilon_{\text{low}}) \in {(0.4,0.1),(0.1,0.4)}$ and **symmetric** $(0.2,0.2)$ clipping settings.
>
> As shown in Table c (see ***https://i.ibb.co/zhWHdwtq/CISPO.png***), **D-ARL consistently outperforms CISPO across all configurations on Qwen3-4B for both mathematical reasoning and code generation tasks**. These results indicate that our conclusions are not sensitive to the specific clipping design and demonstrate the robustness of our method.
>
> ### Weakness 4 & Question 5
> > **Do the authors have any evidence that the method transfers beyond the current Qwen3 plus GRPO-style setting?.**
>
> We sincerely thank the reviewer for the valuable suggestion.  To evaluate generalization of our method, we conduct additional experiments on **DeepSeek-Distill-Qwen-7B**.
>
> As shown in Table d (see ***https://i.ibb.co/PvzJJ8TP/Deepseek-Distill-Qwen-7-B.png***), D-ARL achieves **an average improvement of 7.3% (68.05 vs. 60.75)** over the synchronous GRPO baseline across four reasoning and code generation tasks. These consistent gains on a different model family demonstrate that **our method generalizes beyond the original setting and extends to broader RL post-training scenarios**.
>
> ### Question 4
> > **What is the computational overhead of the replay-buffer selection step relative to the underlying GRPO update?.**
>
> We sincerely appreciate the reviewer for this valuable question. To assess the computational overhead, we measure its time cost relative to sample generation.
>
> As shown in Table e (see ***https://i.ibb.co/Mx2hMjQ6/sample-selection-ratio.png***), the selection step accounts for **13.7%** and **13.3%** of the total time on average for Qwen3-1.7B and Qwen3-4B, respectively, indicating **a moderate and practically acceptable overhead**. Moreover, **Figure 4 in the manuscript shows that the efficiency gains from improved performance outweigh this additional cost**, demonstrating that our method improves performance while maintaining overall training efficiency.
>
> ### Weakness 5 and Limitations
> > **The paper does not include a dedicated limitations section addressing these issues directly.**
>
> We sincerely appreciate the reviewer for pointing out the weaknesses of our paper! In the revised version, **we will include a dedicated Limitations section**.

---

### Official Review · Reviewer_ouCA · 2026-03-14

**Soundness:** 2
**Presentation:** 3
**Significance:** 3
**Originality:** 3
**Overall Recommendation:** 4
**Confidence:** 4

**Summary:**

This paper introduces D-ARL, a framework designed to stabilize asynchronous reinforcement learning for post-training of large language models. To solve unstable optimization and degraded performance, D-ARL maintains a replay buffer of recent policy samples and employs a variance-guided metric to select high-quality data that aligns with the current model.

**Compliance With Llm Reviewing Policy:**

Affirmed.

**Final Justification:**

The authors have addressed my concerns during the rebuttal period.  I kept my already positive score.

**Key Questions For Authors:**

See Weaknesses.

**Limitations:**

No.

**Strengths And Weaknesses:**

**Strengths**

1.	D-ARL is the RL framework to explicitly select distribution-matched data and leverage it for stronger, more stable LLM post-training.

2.	The empirical results show significant gains across diverse reasoning benchmarks.

**Weaknesses**

1.	The main text introduces the replay buffer as storing samples from the "most recent $K$ behavior policies". However, in the experimental setup and Appendix E.1, the buffer length is strictly hardcoded to $K=4$. To ensure the robustness of the proposed framework, it is necessary to provide a sensitivity analysis or ablation study for $K$ (e.g., $K \in \{2, 4, 8, 16\}$).

2.	The evaluation is exclusively conducted on six reasoning and code generation benchmarks, and the metrics primarily focus on reasoning performance (exact-match accuracy) and training efficiency. These domains utilize deterministic, rule-based reward functions. It remains unclear how well D-ARL generalizes to standard alignment or preference tuning tasks where reward signals are derived from learned Reward Models (RMs) that introduce noise and shift over time.

3.	The experiments are limited to Qwen3-1.7B and Qwen3-4B. This makes it hard to judge whether the method remains effective in larger-scale RL post-training settings.


**Minor issues**

1. Typo in Figure 1: In the caption of Figure 1, " fromework" should be corrected to "framework".

---

> ### Author Rebuttal · Authors · 2026-03-31
>
> Dear Reviewer ouCA,
>
> We greatly appreciate your careful reading and constructive feedback! We have provided further responses as follows. We sincerely hope that our responses could properly address all your concerns. If so, we would deeply appreciate it if you could raise your score. If not, please let us know your further concerns, and we will continue actively responding to your comments and improving our submission.
>
> ### Weakness 1
> > **Robustness: To ensure the robustness of the proposed framework, it is necessary to provide a sensitivity analysis or ablation study for the replay buffer size $K$ (e.g., $K \in 2, 4, 8, 16$).**
>
> We sincerely thank the reviewer for this insightful suggestion. **This concern has been addressed in our response to Reviewer ULAG (Weakness 1)**, where we provide a detailed ablation study on the replay buffer size $K$ (e.g., $K \in {2,4,8,16}$) and analyze its effect on **both reasoning performance and computational efficiency**. We kindly refer the reviewer to that section for more details.
>
>
> ### Weakness 2
> > **Reward functions: It remains unclear how well D-ARL generalizes to standard alignment or preference tuning tasks where reward signals are derived from learned Reward Models (RMs) that introduce noise and shift over time.**
>
> We sincerely appreciate the reviewer’s insightful comment. To evaluate the generalization capability of D-ARL in standard alignment tasks with learned reward signals, we further assessed our method on **WritingBench[1]**, which spans multiple diverse domains, including Academic & Engineering, Finance & Business, etc.
>
> For evaluation, we report the **Overall Score**, which represents the comprehensive quality assessment of the generated text. It is calculated by aggregating the model's performance across all domain subsets and specific writing constraints (e.g., style, format, and length) as evaluated by the benchmark's Reward Model.
>
> The Results in Table a show that within the same training steps, D-ARL outperforms the synchronous GRPO baseline, with **an improvement of +0.022 of the Overall Score**. It demonstrates that D-ARL generalizes well to scenarios where reward signals are derived from learned, potentially noisy RMs that are subject to distributional shifts.
>
> **Table a:** Comparison between D-ARL and GRPO in terms of reasoning performance on the WritingBench after one training epoch. We use DeepSeek-Distill-Qwen-7B as the backbone model.
> | Deepseek-Distill-Qwen-7B | WritingBench |
> |--------------------------|---------|
> | Method                   | OverallScores |
> | GRPO                     | 7.092   |
> | D-ARL (Ours)             | **7.114** |
>
> [1] Y. Wu, J. Mei, M. Yan, et al. *WritingBench: A Comprehensive Benchmark for Generative Writing*. NeurIPS 2025 Datasets and Benchmarks Track.
>
> ### Weakness 3
> > **Model Scaling: The experiments are limited to Qwen3-1.7B and Qwen3-4B. This makes it hard to judge whether the method remains effective in larger-scale RL post-training settings.**
>
> We sincerely appreciate the reviewer’s valuable suggestion. To evaluate the effectiveness of our method on larger-scale models, we further conduct experiments using **larger model Qwen3-14B as the backbone**.
>
> The results in Table b demonstrate that **D-ARL achieves average training time reductions of 15.5% and 9.1% compared to the synchronous GRPO baseline**, while maintaining comparable reasoning performance on mathematical reasoning and code generation tasks, respectively. Notably, these gains are consistent with those observed on smaller models, suggesting that our method scales reliably with model size. Furthermore, the stable performance across different task domains indicates that D-ARL maintains robust optimization behavior in larger-scale RL post-training settings. Overall, these results demonstrate that **D-ARL generalizes effectively and remains beneficial as model scale increases**.
>
> **Table b:** The results demonstrate that for Qwen3-14B, our D-ARL outperforms the synchronous GRPO baseline in terms of the training time and reasoning performance.
> | Qwen3-14B | AIME24 |  |  | LiveCodeBench |  |  |
> |-----------|--------|--|--|---------------|--|--|
> | Method | Acc (%) | Time (s) | Impr (Time, %) | Acc (%) | Time (s) | Impr (Time, %) |
> | GRPO | 0.75 | 29.47 | NA | 0.62 | 88.18 | NA |
> | D-ARL (Ours) | **0.75** | **24.9** | **15.5** | **0.64** | **80.14** | **9.1** |
>
> ### Minor issues
> > **Typo in Figure 1: In the caption of Figure 1, " fromework" should be corrected to "framework**.
>
> We sincerely thank the reviewers for pointing out this typo. We will correct “fromework” to “framework” in the revised version.

---

> > ### Author Rebuttal · Reviewer_ouCA · 2026-04-01
> >
> > I thank the authors for their rebuttal. I will keep my already positive score.

---

> > > ### Author Response · Authors · 2026-04-02
> > >
> > > Dear Reviewer ouCA,
> > >
> > > Thank you very much for your time and for reading our rebuttal! We sincerely appreciate your positive feedback and your continued support！
> > >
> > > Best regards,
> > >
> > > Authors of Submission 20518

---

### Official Review · Reviewer_ULAG · 2026-03-18

**Soundness:** 3
**Presentation:** 3
**Significance:** 3
**Originality:** 3
**Overall Recommendation:** 5
**Confidence:** 4

**Summary:**

This paper present DARL, which is a method to alleviate the offpolicy problem for rollout collection in async rl. D-ARL fixes this by selecting samples from a replay buffer of recent policies that best match the current policy's distribution. It then uses theoretically derived weights to aggregate these multi-source samples during optimization.

They evaluated on the following datasets: Math (GSM8K, LightEval, MATH-500, AIME24), and Code (LiveCodeBench, HumanEval).
Results (vs best async baseline CISPO):
Qwen3-1.7B: 65.2% avg vs 58.8% (+6.4%)
Qwen3-4B: 74.3% avg vs 68.7% (+5.6%)
They also reach the same accuracy as CISPO using only 65.3% of the training steps, and 37.2% less wall-clock time.

**Compliance With Llm Reviewing Policy:**

Affirmed.

**Key Questions For Authors:**

Can you present more analysis on the hyper-parameters mentioned in the weaknesses section?

**Limitations:**

yes

**Strengths And Weaknesses:**

# Strengths

Importance sampling for reusing existing data is not new, but DARL chooses only samples with a small ppo ratio, and assign weights to previous K policies in the past. The great experiment results shows the efficiency improvement brought by this approach.

# Weaknesses

While I am really excited about the idea, I want to point out the lack of analysis on several engineering designs in this method. Especially,

1. No ablation on K — what happens with K=2, K=8, K=16?
2. No analysis of what off-policy degree threshold is appropriate — they just take top-k lowest degree samples but don't study sensitivity to k
3. No study of when samples become "too stale" to be useful
4. No analysis of how fast the distribution shifts across training, which would directly inform how large K should be

And, to nitpick a bit more, the field might be more excited about experiments on agentic RL and long-horizon tasks, where async RL is super important for scaling up.

---

> ### Author Rebuttal · Authors · 2026-03-31
>
> Dear Reviewer ULAG,
>
> We greatly appreciate your careful reading and constructive feedback! We have provided further responses as follows. We sincerely hope that our responses could properly address all your concerns. If so, we would deeply appreciate it if you could raise your score. If not, please let us know your further concerns, and we will continue actively responding to your comments and improving our submission.
>
> ### Weakness 1
> > **No ablation on K——what happens with K=2, K=8, K=16?**
>
> We sincerely appreciate the reviewer for the insightful questions. To address this concern, we conducted additional experiments by varying the replay buffer size $K \in \{2, 4, 8, 16\}$.
>
> Specifically, the results in Table a show that, **as the replay buffer size $K$ increases**, **(1) the reasoning performance initially improves and achieves its best performance at $K=4$**, after which it exhibits diminishing or slightly degraded performance for larger $K$; and **(2) the time cost of sample selection correspondingly increases**.
>
> A plausible explanation for this phenomenon is that **when $K$ is too small, the replay buffer contains insufficient high-quality samples**, limiting effective selection; whereas **when $K$ is too large, the buffer introduces excessive noise from highly off-policy samples**, making it more difficult to identify samples that are well aligned with the current policy.
>
> Therefore, we select $K=4$ in practice, as it achieves the best trade-off between reasoning performance and computational efficiency.
>
> **Table a:** The ablation results for the replay buffer size $K$ using Qwen3-4b model. Time refers to the average sample selection time per training step.
> | Qwen3-4B | MATH-500 |  | HumanEval |  |
> |------------------|----------|--|------------|--|
> | **Buffer Size**                | Acc (%)  | Normalized Time | Acc (%) | Normalized Time |
> | $K=2$                | 83.9    | 0.12            | 87.2   | 0.24            |
> | $K=4$                | **87.3**    | 0.23            | **90.2**   | 0.30            |
> | $K=8$                | 84.5    | 0.44            | 89.6   | 0.53            |
> | $K=16$               | 85.5    | 1.00            | 82.9   | 1.00            |
>
> ### Weakness 2
> > **No analysis of what off-policy degree threshold is appropriate.**
>
> We sincerely appreciate the reviewer for the insightful comments. At each training step, a batch of samples will be selected from the replay buffer for policy optimization, and the batch size is fixed by design. Since the distribution of off-policy degrees varies across training, **it is difficult to define a universal selection threshold**. To address this, we adopt a top-$k$ strategy with **$k$ equal to the batch size**, ensuring that we **consistently select the samples with the lowest off-policy degrees** (i.e., those most aligned with the current policy). Compared to threshold-based methods, this approach is more stable and adaptable across different training stages.
>
> ### Weakness 3
> > **No study of when samples become "too stale" to be useful**
>
> We sincerely appreciate the reviewer’s insightful comments. To further investigate the effect of sample staleness, we provide additional training analysis.
>
> As shown in **Figure 6 of the manuscript**, our method achieves a **significantly faster convergence rate** compared to baseline approaches. This suggests that, within our framework, samples stored in the replay buffer may become “stale” more rapidly. Importantly, this phenomenon highlights the effectiveness of our approach: our selection mechanism can **identify and reuse high-quality samples that remain well aligned with the current policy, thereby significantly improving sample efficiency**.
>
> ### Weakness 4
> > **No analysis of how fast the distribution shifts across training, which would directly inform how large K should be**
>
> We sincerely appreciate the reviewer’s valuable suggestion. Following this comment, we conduct an analysis of the distributional shift between the selected samples generated by neighboring policies and the current policy.
>
> Specifically, we vary the replay buffer size $K \in \{2, 4, 8, 16\}$, and measure **the distributional discrepancy between the behavior policy $\pi_q$ and the current policy $\pi_p$ using $|\pi_q(\tau)-\pi_p(\tau)|$**, where $\tau$ denotes the samples selected for training. The figures in the anonymous link ***https://i.ibb.co/1fpB5rkb/distribution-shift.png*** show that **$K=4$ achieves the fastest convergence (10 steps)**, compared to 15, 21, and 20 steps for $K=2,8,16$, respectively. **This trend is consistent with the trade-off discussed in Weakness 1**: $K=4$ strikes the best balance between sample diversity and off-policy noise, leading to faster distributional alignment.
>
> ### Question
> > **Can you present more analysis on the hyper-parameters?**
>
> We have provided a detailed analysis of the hyperparameters in the Weakness section. Please refer to our responses to Weaknesses 1–4 for further details.

---

> > ### Author Rebuttal · Reviewer_ULAG · 2026-04-04
> >
> > I thank the authors' response which resolves my concerns. Please include these in the camera ready.

---

> > > ### Author Response · Authors · 2026-04-07
> > >
> > > Dear Reviewer ULAG,
> > >
> > > Thank you very much for your time and for reading our rebuttal! We sincerely appreciate your positive feedback and your continued support! We will include the modifications in the camera ready.
> > >
> > > Best regards,
> > >
> > > Authors of Submission 20518

---

### Decision · Program_Chairs · 2026-04-30

**Decision:**

Accept (regular)

**Comment:**

This paper proposes D-ARL, a distribution-matched asynchronous RL framework for LLM post-training that selects replay-buffer samples well-aligned with the current policy via a variance-guided metric, then applies a multi-behavior policy optimization step. Reviewer scores are 5, 5, 4, and 3. ULAG (5) was impressed by the empirical efficiency gains from selective importance sampling across replay policies. 9vbQ (5, raised from 4) found the paper well-organized and focused, with most design choices supported by clear theoretical guarantees, and called the performance-vs-compute tradeoff "genuinely surprising." ouCA (4) highlighted the practical significance of explicitly selecting distribution-matched data and found the gains across diverse benchmarks meaningful. The sole dissenting reviewer, f3gZ (3), raised concerns about the theory-to-implementation gap in Theorem 4.1 as well as limited diversity in RL objectives tested and an unanswered question about the Decoupled baseline configuration. The rebuttal resolved several of f3gZ's concerns (multi-seed robustness, CISPO sensitivity, computational overhead), as acknowledged explicitly by the reviewer, but the core theoretical gap was not fully addressed and f3gZ did not update their score.

On balance, with three reviewers at 4–5 and strong empirical evidence across six benchmarks and multiple model sizes, the contribution is solid enough for acceptance. The remaining open points from f3gZ are legitimate and should be addressed in the camera-ready, particularly the limitations section, a clearer discussion of the theory-practice gap, and the Decoupled baseline setup. I recommend Accept.